# ON ROBUST PREFIX-TUNING FOR TEXT CLASSIFICATION

**Zonghan Yang, Yang Liu**
Department of Computer Science and Technology, Institute for AI Industry Research
Institute for Artificial Intelligence, Tsinghua University, Beijing, 100084, China
`yangzh20@mails.tsinghua.edu.cn, liuyang2011@tsinghua.edu.cn`

## ABSTRACT

Recently, prefix-tuning has gained increasing attention as a parameter-efficient finetuning method for large-scale pretrained language models. The method keeps the pretrained models fixed and only updates the prefix token parameters for each downstream task. Despite being lightweight and modular, prefix-tuning still lacks robustness to textual adversarial attacks. However, most currently developed defense techniques necessitate auxiliary model update and storage, which inevitably hamper the modularity and low storage of prefix-tuning. In this work, we propose a robust prefix-tuning framework that preserves the efficiency and modularity of prefix-tuning. The core idea of our framework is leveraging the layerwise activations of the language model by correctly-classified training data as the standard for additional prefix finetuning. During the test phase, an extra batch-level prefix is tuned for each batch and added to the original prefix for robustness enhancement. Extensive experiments on three text classification benchmarks show that our framework substantially improves robustness over several strong baselines against five textual attacks of different types while maintaining comparable accuracy on clean texts. We also interpret our robust prefix-tuning framework from the optimal control perspective and pose several directions for future research [1].

## 1 INTRODUCTION

Large-scale pretrained language models (LMs) (Peters et al., 2018; Devlin et al., 2019; Radford et al., 2019; Liu et al., 2019; Yang et al., 2019; Raffel et al., 2020; Lewis et al., 2020; Brown et al., 2020; Xue et al., 2021) have proven effective for downstream NLP tasks. While finetuning a pretrained model for a specific task has been the common practice, it comes at the cost of maintaining a full copy of the LM with the parameters entirely modified. The prohibitively huge memory demand poses a severe challenge for the deployment of practical NLP systems, which motivates the development of *low-storage adaptation* methods (Houlsby et al., 2019; Li & Liang, 2021).

Recently, increasing interest has been focused on prompt-based tuning approaches for pretrained language models (Wallace et al., 2019; Puri & Catanzaro, 2019; Shin et al., 2020; Jiang et al., 2020b; Zhong et al., 2021; Gao et al., 2021; Hu et al., 2021; Liu et al., 2021). By prepending several elaborately-selected tokens to the given input sequences, the LM is triggered to respond with appropriate outputs without updating its parameters. Prefix-tuning (Li & Liang, 2021) introduces the idea of replacing the discrete prompt tokens at the input with the virtual ones at the start of each layer in the LM. By optimizing the layerwise continuous prefix embedding instead of selecting candidates in the vocabulary list, the expressive ability of prompts is further enhanced with a rather small amount of parameters to be updated. As a result, prefix-tuning requires near $1000\times$ fewer task-specific parameters than finetuning the entire pretrained model (Bommasani et al., 2021).

Despite being lightweight and modular, prefix-tuning is still lacking in robustness. In the NLP community, a variety of techniques for generating adversarial examples have been proposed to attack a text classifier by perturbing inputs (Zhang et al., 2020). Conventional attack techniques include character-level (Eger et al., 2019; He et al., 2021), word-level (Alzantot et al., 2018; Ren et al.,

---

[1] We release the code at `https://github.com/minicheshire/Robust-Prefix-Tuning`

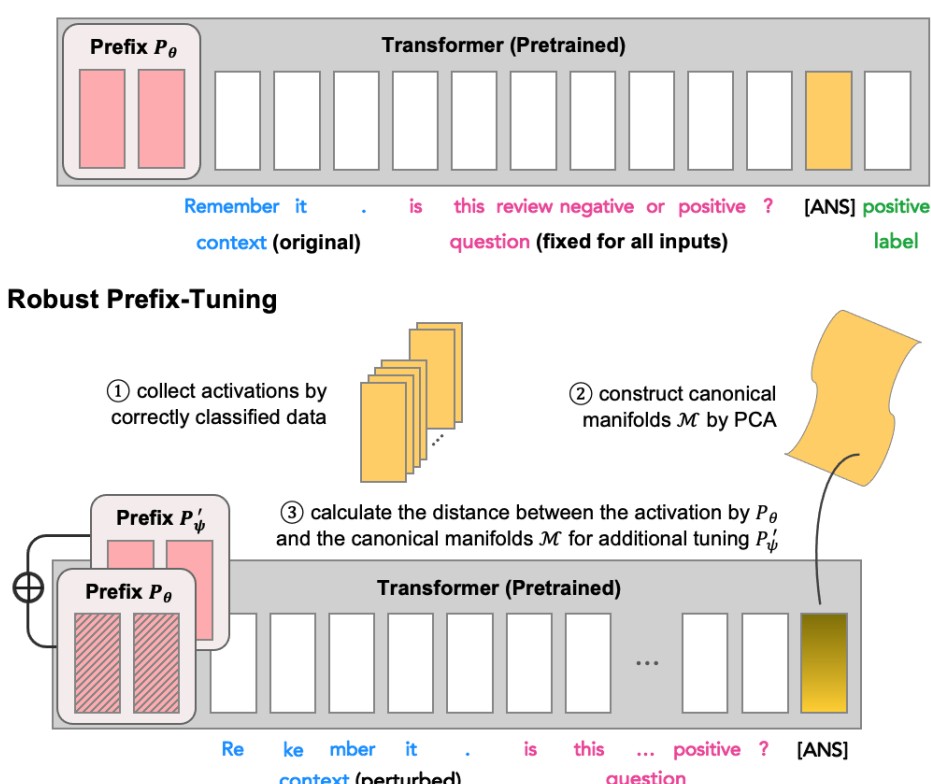

Figure 1: Overview of prefix-tuning as well as our robust prefix-tuning framework for text classification. We frame the samples into a SQuAD-like scheme consisting of context, question, and label and optimize the original prefix $P_\theta$. For the robust prefix-tuning framework, we fix the obtained prefix $P_\theta$ and tune an additional prefix $P'_\psi$ for each test batch. The additional tuning follows the three steps indicated in the figure, which aims to lead the summed prefix to steer correct activations at the position of the [ANS] token with those activated by correctly classified training data as the standard.

2019; Garg & Ramakrishnan, 2020), sentence-level modification (Iyyer et al., 2018; Ribeiro et al., 2018; Xu et al., 2021), or a mixture of them (Ebrahimi et al., 2018; Li et al., 2019). Instead of perturbing each input sentence separately, recently, universal adversarial triggers (UAT) (Wallace et al., 2019) becomes powerful by prepending the same adversarial tokens to all test inputs. UAT prompts the model to generate malicious outputs, which shares the same spirit with the prompt-based tuning approaches. It remains a mystery whether prefix-tuning, a variant of prompt-based tuning techniques, can defend against UAT as well as other different kinds of attacking techniques.

In defense of adversarial attacks, different types of defense techniques are developed, including model functional improvement (Li & Sethy, 2019; Jones et al., 2020), certification (Jia et al., 2019; Huang et al., 2019; Shi et al., 2020; Xu et al., 2020; Ye et al., 2020), adversary detection (Pruthi et al., 2019; Zhou et al., 2019), and adversarial training (Miyato et al., 2017; 2019; Zhu et al., 2020; Jiang et al., 2020a; Liu et al., 2020; Wang et al., 2021; Dong et al., 2021; Zhou et al., 2021). While these approaches have enhanced model robustness, difficulties emerge when fitted to prefix-tuning. Most of the techniques require modification to the architecture and the parameters of the LM or additional maintenance of adversary detectors. Directly applying such techniques necessitates auxiliary model update and storage, which will inevitably hamper the modularity of prefix-tuning. Moreover, The excessively long time for adversarial training is also a hindrance to the efficient use of prefix-tuning. We ask the following question: *Can we improve the robustness of prefix-tuning while preserving its efficiency and modularity, without modifying the pretrained model parameters?*

In this work, we propose a robust prefix-tuning framework for text classification. The main idea of our framework is to add an extra batch-level prefix tuned for each batch to the original prefix embedding during test time for robustness enhancement. We first record the layerwise activations in the

LM at the position of generating label prediction with correctly classified training data. We project the collected activation matrices of each layer onto low-level canonical manifolds as the characterization of "correct" model behavior. In this way, the correctness of any layerwise activations at the position of prediction generation can be estimated by projecting to the canonical manifolds and measuring the distance between them. For each test batch during inference, the added extra prefix is tuned on the fly with the original prefix fixed to minimize the calculated distance. Triggered by the summed prefix, the LM is prone to generating correct label predictions. We conduct extensive experiments on three text classification benchmarks and show that the proposed framework substantially improves model robustness against five strong textual attack approaches including input perturbation attack of different levels as well as the UAT attack. To the best of our knowledge, we are the first to propose the defense approach for prefix-tuning while keeping its lightweightness and modularity. Moreover, we provide an interpretation of our robust prefix-tuning framework from the optimal control perspective and pose several directions for future research.

## 2 PREFIX-TUNING FOR TEXT CLASSIFICATION

Prefix-tuning is a lightweight alternative to finetuning when using large-scale pretrained language models to solve downstream NLP tasks. The intuition of prefix-tuning follows prompt-based methods that a proper context prepended to input sentences triggers the desired response of the LM without changing the large amount of LM parameters. Instead of instantiating the prepended context with discrete tokens, prefix-tuning uses trainable prefix embeddings as a replacement, which is also known as *soft prompts*. The continuous prefix embeddings enable continuous optimization and are prepended to all Transformer layers to improve expressiveness. Following the notation of Li & Liang (2021), the activation at the $i$-th position of the $j$-th layer in an $L$-layer autoregressive Transformer LM is denoted as $h_i^{(j)}$. $h_i = [h_i^{(0)}; \cdots; h_i^{(L-1)}]$ represents the stacked activations:

$$h_i = \begin{cases} P_\theta[i, :], & \text{if } i \in \mathrm{P_{idx}}, \\ \mathrm{LM}_\phi(z_i, h_{<i}), & \text{otherwise.} \end{cases} \tag{1}$$

where $\mathrm{P_{idx}}$ is the sequence of prefix indices and $z_i$ is the $i$-th token in the input sequence. The activations of the first $|\mathrm{P_{idx}}|$ positions are directly calculated by $P_\theta$. All of the activations at the following positions depend on the prefix as the autoregressive LM follows the left-to-right calculation process. To stabilize the optimization, the prefix embedding matrix $P_\theta$ is reparameterized as $P_\theta[i, :] = \mathrm{MLP}_\theta(\hat{P}_\theta[i, :])$ by a feedforward network $\mathrm{MLP}_\theta$ with a smaller matrix $\hat{P}_\theta$.

While prefix-tuning is proposed for conditional generation tasks, in this work, we use prefix-tuning for text classification. As shown in Figure 1, following the protocol of decaNLP (McCann et al., 2018), we frame the samples in classification tasks into a SQuAD-like scheme consisting of context, question, and label: the context and the label part refer to the text sequence to be classified and the ground-truth label, while the question part is a prescribed task description sentence fixed for all samples. We denote $x = [\text{context}, \text{question}, [\text{ANS}]]$, where $[\text{ANS}]$ is a special token that separates question and label. We let $y = [\text{label}]$ and $|y| = 1$ as the label is one token. At the position that $[\text{ANS}]$ is inputted, the LM generates the prediction of the next label token, and we denote this position as the output position $o$. While $o$ can be different for different input $x$'s, in this paper, we omit the relation $o = o(x)$ for simplicity. Prefix-tuning aims to steer the LM to maximize the probability of the label. We use all samples in the training set $\mathcal{D}_{tr}$ to optimize the prefix $P_\theta[i, :]$. The objective is

$$\min_\theta \mathbb{E}_{(x,y) \sim \mathcal{D}_{tr}} \mathcal{L}(y|x; \theta) = \max_\theta \mathbb{E}_{(x,y) \sim \mathcal{D}_{tr}} \log \left[ W\left(h_o^{(L)}\right) \right]_y, \tag{2}$$

where $W$ in the LM transforms the top-layer output $h_o^{(L)}$ to a probability vector over the vocabulary.

With continuous optimization on training samples, prefix-tuning is expected to steer the LM to generate correct label predictions for test data. With the large-scale LM parameters fixed, the obtained task-specific prefix is lightweight and modular. However, prefix-tuning is still vulnerable to text attacks. With the context part perturbed by text attack techniques, the LM can be fooled to generate erroneous label prediction at the output position. Figure 1 shows an example of perturbation: by modifying a single character $m$ in the word *remember* with $k$, the prediction of the LM is shifted from positive to negative. Therefore, it remains under exploration how to robustify prefix-tuning without hampering its modularity or introducing additional large model updates and storage.

## 3 ROBUST PREFIX-TUNING

We propose a robust prefix-tuning framework for text classification. Our intuition follows prefix-tuning that proper prefix embeddings prepended to inputs can steer a LM with correct responses. When the inputs are adversarially perturbed, the LM activations at the output position fail to be steered in the correct way by the original prefix $P_\theta[i,:]$. Inspired by Khoury & Hadfield-Menell (2018) that the perturbed data often deviates from the low-dimensional data manifold, our robust prefix-tuning framework uses the layerwise activations by correctly classified training data to construct canonical manifolds $\mathcal{M}$. When provided with perturbed inputs during inference, we add an extra prefix $P'_\psi[i,:]$ tuned for each test batch to $P_\theta[i,:]$ that aims to rectify the erroneous activations at the output position so that they stay close to the canonical manifolds. In this way, we expect the summed prefix to steer the LM with correct label generation against input perturbations. As shown in Figure 1, our robust prefix-tuning framework consists of three steps. The first step is collecting correct LM activations at the output position $o$ triggered by $P_\theta[i,:]$. We denote $S_C$ as the set of correctly classified training examples. For the $j$-th layer, the collected activation matrix $H_C^{(j)}$ stacks the $j$-th layer LM activation at the output position $o$ with the input of all $c \in S_C$:

$$H_C^{(j)} = \left[ h_{o,c}^{(j)} \right] \in \mathbb{R}^{|S_C| \times d}. \tag{3}$$

The $d$ represents the dimension of the LM hidden state. In practice, we always have $|S_C| >> d$.

The second step is constructing canonical manifolds. We project the collected $j$-th layer activation matrix $H_C^{(j)}$ onto a low-level manifold $\mathcal{M}^{(j)}$ as the characterization of the correct $j$-th layer behavior. We use PCA (Pearson, 1901) to get the projection $Q^{(j)}$ onto the canonical manifold of the $j$-th layer:

$$\widetilde{H}_C^{(j)} = U^{(j)} \Sigma^{(j)} V^{(j)\mathrm{T}}, \tag{4}$$

$$Q^{(j)} = V_p^{(j)\mathrm{T}} V_p^{(j)}, \tag{5}$$

where $\widetilde{H}_C^{(j)} = H_C^{(j)} - \mathbf{1}\mathbf{1}^{\mathrm{T}} H_C^{(j)} / |S_C|$ normalizes the rows of $H_C^{(j)}$ to mitigate the randomness among samples in $S_C$ before projection. $V_p^{(j)}$ consists of the first $p$ singular vectors and $Q^{(j)} \in \mathbb{R}^{d \times d}$.

The third step is tuning $P'_\psi[i,:]$ to robustify prefix-tuning during inference. Here the vector $P'_\psi[i,:]$ is *not* reparameterized by MLP. With the additional prefix $P'_\psi[i,:]$, the token-wise activations become

$$h_i = \begin{cases} P_\theta[i,:] + P'_\psi[i,:], & \text{if } i \in \mathrm{P_{idx}}, \\ \mathrm{LM}_\phi(z_i, h_{<i}), & \text{otherwise.} \end{cases} \tag{6}$$

For the $j$-th layer at the output position, the LM activation matrix triggered by $P_\theta[i,:] + P'_\psi[i,:]$ with the potentially perturbed test input batch $S_T$ is stacked as

$$H_T^{(j)} = \left[ h_{o,t}^{(j)} \right] \in \mathbb{R}^{|S_T| \times d} \tag{7}$$

for all $t \in S_T$. We use the distance from $H_T^{(j)}$ to the $j$-th canonical manifold $\mathcal{M}^{(j)}$ as the loss for the tuning of $P'_\psi[i,:]$ for each batch. Projecting $H_T^{(j)}$ to $\mathcal{M}^{(j)}$ yields $H_T^{(j)} Q^{(j)}$, thus the objective is

$$\min_\psi \sum_{j=0}^{N-1} L\left( \psi^{(j)}, Q^{(j)} \right) = \sum_{j=0}^{N-1} \left\| H_T^{(j)} \left( I - Q^{(j)} \right) \right\|_2. \tag{8}$$

We also replace the $H_T^{(j)}$ in Eq. (8) with $H_T^{(j)} - \mathbf{1}\mathbf{1}^{\mathrm{T}} H_T^{(j)} / |S_T|$ as normalization before projection to mitigate randomness among test samples when $|S_T| > 1$. After tuning $P'_\psi[i,:]$, the activated $H_T^{(j)}$ is closer to $\mathcal{M}^{(j)}$. As the manifold characterizes the correct behavior of the $j$-th layer activation, by regulating the layerwise activations at the output position, the summed prefix $P_\theta[i,:] + P'_\psi[i,:]$ is prone to steering the LM to generate correct label predictions. Our framework is also applicable to other soft prompt-based tuning methods (Qin & Eisner, 2021; Hambardzumyan et al., 2021; Lester et al., 2021; Cho et al., 2021; Tsimpoukelli et al., 2021) by recording the activations of correctly classified training data, constructing canonical manifolds for the soft prompts, and tuning additional soft prompts for robustness during inference. In this work, we conduct experiments on prefix-tuning.

**Remark.** *From the optimal control (OC) perspective, prefix-tuning can be formalized as seeking the OC of the pretrained LM for downstream tasks, and our robust prefix-tuning can be interpreted as seeking the close-loop control for robust downstream tasks. We attach the details in Appendix G.*

Table 1: Results of different baselines with our framework applied. Our framework substantially improves robustness of both standard and adversarial prefix-tuning against all types of attacks.

| Benchmark | Method | Clean | PWWS | VIPER | SCPN | BUG | UAT |
|---|---|---|---|---|---|---|---|
| SST-2 | std. prefix-tuning | 92.48 | 16.64 | 1.92 | 31.58 | 8.84 | 5.05 |
| | + our framework | **92.59** | **50.36** | **44.65** | **58.54** | **46.68** | **85.72** |
| | adv. prefix-tuning | 93.57 | 30.64 | 7.25 | 35.42 | 25.04 | 4.88 |
| | + our framework | **93.79** | **57.55** | **43.60** | **60.68** | **57.17** | **91.87** |
| AG's News | std. prefix-tuning | 86.42 | 43.00 | 24.55 | 43.20 | 43.22 | 52.20 |
| | + our framework | **86.50** | **53.91** | **24.93** | **48.38** | **51.79** | **76.47** |
| | adv. prefix-tuning | 89.46 | 50.91 | 26.30 | 45.25 | 47.11 | 59.74 |
| | + our framework | **90.26** | **56.45** | **31.25** | **48.03** | **54.66** | **87.26** |
| SNLI | std. prefix-tuning | 72.48 | 25.51 | 29.69 | 42.46 | 32.88 | 30.20 |
| | + our framework | **72.74** | **34.68** | **33.64** | **43.37** | **36.59** | **71.03** |
| | adv. prefix-tuning | 77.22 | 27.43 | 28.58 | 46.83 | 34.94 | 35.76 |
| | + our framework | **77.65** | **33.88** | **33.98** | **46.92** | **38.21** | **76.15** |

## 4 EXPERIMENTS

### 4.1 SETUP

While we have also provided the regular finetuning as baselines We consider three text classification benchmarks in our experiments: binary Stanford Sentiment Treebank (SST-2) (Socher et al., 2013), AG's News (Zhang et al., 2015), and Stanford Natural Language Inference (SNLI) (Bowman et al., 2015). We evaluate our robust prefix-tuning with five text attacks: PWWS (Ren et al., 2019), VIPER (Eger et al., 2019), SCPN (Iyyer et al., 2018), TextBugger ("BUG" for short) (Li et al., 2019), and UAT (Wallace et al., 2019). We use the GPT2-medium (with $L = 24$ layers in total) as the large-scale LM and set prefix length $= 10$ for all prefix-tuning experiments. We train 100 epochs for SST-2 and 25 epochs for AG's News and SNLI. We set $N = 3$ and record the bottom $N$-layer activations of the LM at the output position for the additional tuning. Other experiment configurations and details can be found in Appendix B. We have also conducted both standard and adversarial full tuning experiments (Appendix A) to discuss the challenges and opportunities in robustifying prefix-tuning.

### 4.2 RESULTS ON ADVERSARIAL TRAINING

We first apply adversarial training to prefix-tuning as our baselines named adversarial prefix-tuning. Following Miyato et al. (2017), we inject perturbation restricted within the $\ell_2$ ball of the flattened word embedding of original sentences during training to adversarially optimize the prefix $P_\theta[i, :]$. We have also attempted to use other types of adversarial training, (i.e., adding the KL-divergence term for regularization (Miyato et al., 2019; Zhang et al., 2019)) but obtained poor accuracy, suggesting the difficulty of optimizing the small amount of prefix embedding parameters. The experimental details can be found in Appendix C.1. Our framework is applicable to the adversarially-trained prefix embedding, as it keeps the acquired prefix $P_\theta[i, :]$ fixed and tunes extra $P'_\psi[i, :]$ for each test batch. The experimental results are listed in Table 1. According to Table 1, our approach significantly improves the robustness of prefix-tuning over all baselines against each type of text attack. To the best of our knowledge, our framework is the first to defend against UAT on large-scale pre-trained LMs (see Appendix A for details). Compared with the standard prefix-tuning baseline, the adversarial baselines achieve better robustness and clean accuracy. In fact, fewer training epochs are needed to achieve certain clean accuracy when applying adversarial training to prefix-tuning. Figure 2-(a) illustrates the loss on the training set and the clean accuracy on the validation set for each epoch during standard and adversarial prefix-tuning. From Figure 2-(a), it is clear that adversarial prefix-tuning outperforms standard prefix-tuning in both the convergence rate and generalization.

However, the adversarial prefix-tuning approaches suffer from the catastrophic drawback of taking far greater training time than the standard prefix-tuning baseline. Figure 2-(b) changes the horizontal axis in Figure 2-(a) from epoch to clock time. According to Figure 2-(b), given the same time budget, standard prefix-tuning finishes training for 100 epochs while adversarial prefix-tuning only manages to train for 20 epochs. The training losses of the two methods at the time are also roughly

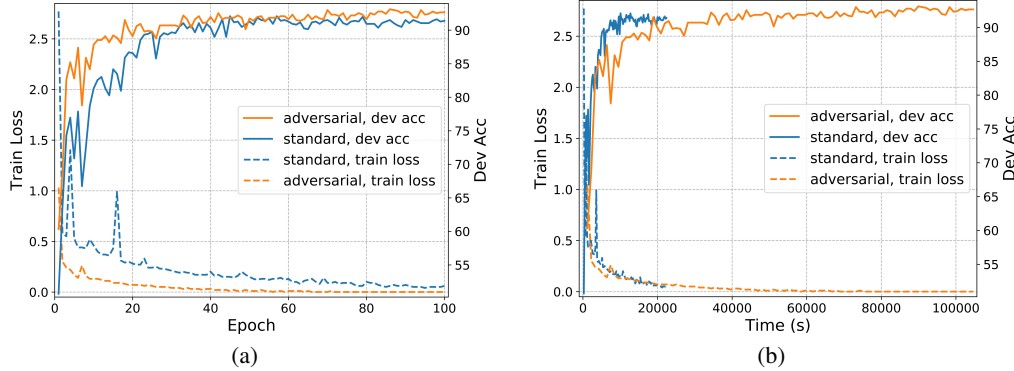

Figure 2: Comparison between standard and adversarial prefix-tuning for 100 epochs with respect to (a) epoch and (b) clock time. While adversarial prefix-tuning gains strengths in epoch-wise convergence rate and generalization, it takes far greater training time than standard prefix-tuning.

the same (0.06). Earlystopping adversarial prefix-tuning at 20 epochs achieves slightly better robustness results than 100 epochs, but the clean accuracy becomes lower than standard prefix-tuning (results can be found in Appendix C.2). To conclude, the adversarial prefix-tuning approaches make regularization effect on prefix-tuning but lack competitiveness in practice.

In contrast with adversarial prefix-tuning, our approach is more practical as the duration of the inference-phase on-the-fly optimization of the additional prefix $P'_\psi[i, :]$ is negligible compared with training-phase adversarial prefix-tuning. From Table 1, the robustness of the std. prefix-tuning applied with our approach has surpassed that of the adv. prefix-tuning without our approach. The adv. prefix-tuning with our framework achieves the strongest robustness against most attacks.

## 4.3 RESULTS ON ADVERSARIAL DATA AUGMENTATION

Table 2: Results of PWWS adversarial data augmentation baselines as well as our methods on the SST-2 benchmark. Our methods consistently improve robustness over all baselines.

| Method | Clean | PWWS | VIPER | SCPN | BUG | UAT |
|---|---|---|---|---|---|---|
| std. prefix-tuning | 92.48 | 16.64 | 1.92 | 31.58 | 8.84 | 5.05 |
| + our framework | **92.59** | **50.36** | **44.65** | **58.54** | **46.68** | **85.72** |
| 20-epoch adv. aug. | **87.15** | 48.11 | 42.12 | 48.11 | 47.01 | 10.43 |
| + our framework | 86.93 | **55.57** | **51.62** | **57.33** | **52.00** | **70.07** |
| 50-epoch adv. aug. | **86.82** | 55.08 | 35.04 | 44.65 | 49.37 | 3.13 |
| + our framework | 86.66 | **58.26** | **48.05** | **58.98** | **51.73** | **83.47** |
| 100-epoch adv. aug. | **91.21** | 56.62 | 23.01 | 38.55 | 45.14 | 7.91 |
| + our framework | 90.39 | **62.55** | **46.95** | **59.25** | **54.75** | **86.38** |

In this section, we conduct adversarial data augmentation during the training phase of standard prefix-tuning as our baseline methods. During training, we use a specified text attack method to perturb each batch and augment the perturbed data to the training set. Due to the computational inefficiency of discrete optimization for text attacks, the training of adversarial data augmentation is even far slower than that of adversarial prefix-tuning introduced in Section 4.2. We set milestones for the training of adversarial data augmentation at 20, 50, and 100 epochs for comparison. When applying our framework, we calculate the canonical manifolds $\mathcal{M}$ using the correctly classified samples from both the clean training set and the perturbed training set by the specified text attack.

Considering the training time budget, we select SST-2 as the benchmark. We use PWWS as the attack method to generate perturbed inputs with results listed in Table 2. On one hand, the clean accuracy of the PWWS adversarial data augmentation method is affected at the 20-epoch and the 50-epoch milestones. This might be attributed to the consideration of perturbed inputs during training. The robustness results, on the other hand, are improved compared with standard prefix-tuning. The robustness against PWWS obtains steady improvement thanks to the adversarial data augmentation

generated by the attack method of the same type. However, for other types of attack, the robustness improvement is decreasing with the training process, suggesting the possibility that overfitting to the specified type of adversarial data can be harmful to robustness against other types of attacks.

With our robust prefix-tuning framework applied, all milestones of the adversarial data augmentation baseline obtain substantial robustness improvement. While we calculate the layerwise canonical manifolds with the correctly classified samples from both the clean training set and the PWWS-perturbed training set, it can be seen that the robustness against PWWS attack is still enhanced by several percent, while the clean accuracy is slightly affected. This shows that when calculating the layerwise manifolds, the used samples from the two types of training data, though are all correctly classified, trigger the LM with activations of different properties at the output position. It is left for future work about how to construct the dataset to calculate the canonical manifolds to achieve the best performance on both accuracy and robustness. We have also used VIPER and SCPN as the attack methods for adversarial data generation and draw similar conclusions to the PWWS adversarial data generation experiments, results of which can be found in Appendix D.

### 4.4 EFFECT OF TUNING DIFFERENT LAYERS

In this section, we study the effect of tuning our robust prefixes with different layers. While we tuned the robust prefixes of the bottom $N = 3$ layers for all previously reported experiments, different results are obtained when tuning the robust prefixes of alternative layers. We experiment on the SST-2 development set by tuning the bottom $N$-layer robust prefixes as well as the top ones with $N$ enumerating from 0 to 24 with the step size $= 3$. $N = 0$ represents the original standard prefix-tuning method, and $N = 24$ means tuning all layers. The results are shown in Figure 3. According to the results, tuning the bottom $N$ layers achieves better robustness improvement than

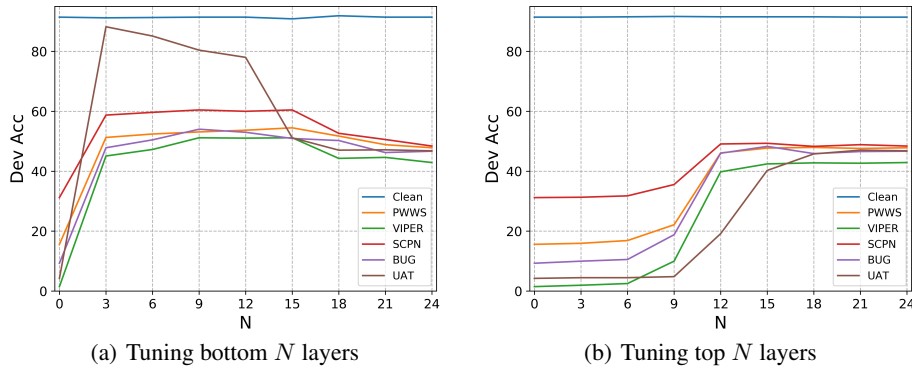

(a) Tuning bottom $N$ layers          (b) Tuning top $N$ layers

Figure 3: Results on the SST-2 development set when applying robust prefixes of different layers. Tuning bottom $N$ layers outperforms tuning top $N$ layers in terms of robustness improvement.

tuning the top ones. This can be attributed to layerwise bottom-up error accumulation of the output position activations triggered with perturbed inputs. For the in-sentence text attacks, tuning the bottom $N \leq 15$ layers achieves comparable robustness improvement and slightly outperforms the larger $N$'s. For the UAT attack, setting the bottom $N = 3$ is significantly better than the choices of larger $N$'s. One possible reason for the exceptional performance is that the collected low-layer activations capture richer positional information (Voita et al., 2019), which helps to defend against UAT as UAT shifts right all inputs with the same trigger. We proceed to study how our method steers the LM with different $N$'s for a better understanding of its robustness improvement in Section 5.

### 4.5 PERFORMANCE UNDER MORE REALISTIC THREAT MODELS

In the previous experiments, the robustness evaluation metric for each method is the accuracy under specific attack on the test set. While this has been adopted in previous work on adversarial defense in NLP (Dong et al., 2021; Si et al., 2021; Li et al., 2021), in this section, we consider more realistic threat models. We first study the effect of test batch size on the performance of our framework. We used an adaptive test batch size to make full use of GPU memory in previous experiments (detailed in Appendix B). However, the loss in Eq. (8) depends on the size of the inference batch, and models in deployment should respond with real-time data. Based on standard prefix-tuning on SST-2, we

Table 3: Performance of our method with std. prefix-tuning with various test batch sizes on SST-2.

| Method | Clean | PWWS | VIPER | SCPN | BUG | UAT |
|---|---|---|---|---|---|---|
| std. prefix-tuning | 92.48 | 16.64 | 1.92 | 31.58 | 8.84 | 5.05 |
| ours, bsz adaptive | 92.59 | 50.36 | 44.65 | 58.54 | 46.68 | 85.72 |
| ours, bsz $= 1$ | **92.64** | **51.46** | 43.99 | 58.32 | 46.73 | 73.09 |
| ours, bsz $= 2$ | 92.48 | 49.92 | 45.74 | **59.14** | 48.93 | 84.73 |
| ours, bsz $= 4$ | **92.64** | 50.36 | **45.85** | 58.54 | **49.31** | **86.66** |

additionally evaluate our framework with small test batch sizes: 1, 2, and 4. According to the results in Table 3, a fixed small test batch size achieves comparable or slightly better results under clean data and in-sentence attacks. Under UAT, however, there exists a performance gap of the robustness between our framework with test batch size of 1 and others. We discuss the phenomenon in depth and attach our attempt for improvement in Appendix E.1). Nevertheless, our framework with test batch size of 1 still substantially outperforms the baseline, and we also find the performance gap is smaller on other benchmarks with other prefix-tuning baselines (see Appendix E.1).

We continue to consider a more challenging yet more realistic setting where the test data is mixed with unperturbed and perturbed samples or perturbed samples under different attacks. We use test batch size of 1 in our framework based on adversarial prefix-tuning for SST-2. While it is unknown under which attack each test sample is perturbed (or whether it is perturbed), we find the results shown in Table 4 still impressive when using a fixed learning rate tuned on the development set under UAT for the additional tuning of $P'_\psi$. We attach the detailed analysis in Appendix E.2.

Table 4: Performance of our framework under mixed test data on SST-2. The '+' denotes combination of test set clean or under attack, and 'C' is short for "Clean", 'B' for "BUG", etc.

| Method | C + B | C + P | V + B | V + P | S + B | S + P | U + B | U + P |
|---|---|---|---|---|---|---|---|---|
| adv. prefix-tuning | 60.65 | 62.27 | 17.76 | 20.43 | 29.57 | 31.85 | 18.12 | 20.81 |
| + ours, bsz $= 1$ | **69.71** | **71.11** | **46.02** | **47.23** | **55.63** | **56.92** | **70.04** | **71.67** |

## 5 HOW DOES ROBUST PREFIX-TUNING STEER THE LM?

As the robust prefix-tuning framework with the proper choice of $N$ substantially improves the robustness of prefix-tuning, it deserves to be explored how the robust prefix-tuning steers the LM compared with the original prefix. In this section, we leverage different proxies to study the behavior of our framework with the standard prefix-tuning baseline. We find that our robust prefix-tuning framework steers different LM behavior between the in-sentence attacks and UAT. For the in-sentence attacks, the behavior of the final layer of the LM can be summarized as *averaging the attention*. For UAT, with the proper setting of $N$, the LM functions as *ignoring the distraction*.

Table 5: Inputs from the SST-2 dev set and their perturbed versions by BUG/UAT attack under which the prediction is flipped. The two examples are used as case studies in Sections 5.1 and 5.2.

| Type | Input (the context part) | Predict | Label |
|---|---|---|---|
| Original | one from the heart . | positive | positive |
| BUG-attack | One from te hart . | negative | positive |
| Original | it 's just filler . | negative | negative |
| UAT-attack | lifts mates who it 's just filler . | positive | negative |

### 5.1 IN-SENTENCE ATTACKS: AVERAGING THE ATTENTION

In this subsection, we leverage attention weights in the final layer of the LM as the proxy and use the adversarial inputs generated by the BUG attack on SST-2 development set for our following in-sentence case studies. We ignore the prefix embeddings and renormalize the attention weights of the rest of the tokens in visualization. Note that the observed behavior is *of the final layer only*: the attention is shown to be averaged only over the vectors inputted to the final layer, as attention is unreliable for indicating importance over input tokens (Serrano & Smith, 2019; Jain & Wallace,

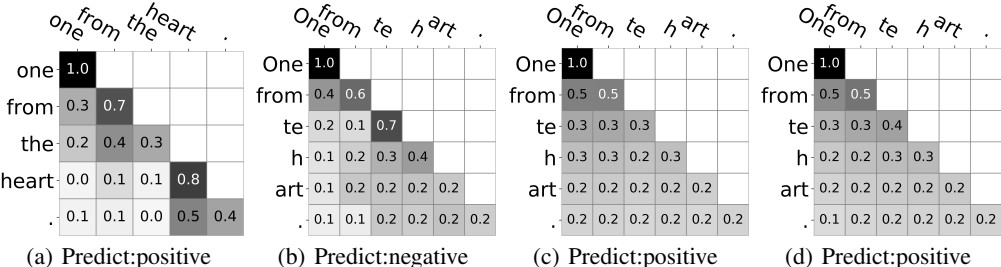

(a) Predict:positive  (b) Predict:negative  (c) Predict:positive  (d) Predict:positive

Figure 4: Visualized attention weight maps of the final layer in the LM for the case study with BUG attack. Each row represents the time step at which the token (labeled on the left) is inputted. Each column in the row illustrates the attention weight assigned to the specific token (labeled on the top). (a): the original input by the original prefix-tuning; (b): the BUG-perturbed input by the original prefix-tuning. Compared with (a), the attention weights are perplexed in (b) due to the perturbed input. (c) and (d): the BUG-perturbed input by our robust prefix-tuning with (c) $N = 24$ and (d) $N = 3$. Our robust prefix-tuning (with both $N = 24$ and $N = 3$) steers the behavior of the final layer to *average the attention* over the input vectors of the final layer (**not** the input tokens).

2019; Wiegreffe & Pinter, 2019; Abnar & Zuidema, 2020). Table 5 shows an example of BUG-perturbed input, and Figure 4 shows the attention map visualization. Similar behavior is observed as well by other in-sentence attacks; more visualization can be found in Appendix F.1.

## 5.2 Universal Adversarial Triggers: Ignoring the Distraction

In this subsection, we leverage an alternative proxy based on the product of gradient and attention as token importance rankings (detailed in Appendix F.2.1). While the proxy is not a gold-standard indicator of importance value either, it can provide more reliable predictions of token importance orderings (Serrano & Smith, 2019). We use UAT to attack SST-2 development set, and Table 5 shows an example of input with triggers "lifts mates who" with visualization in Figure 5. As a result, the

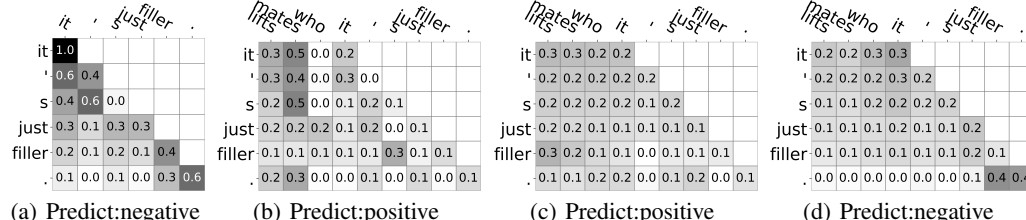

(a) Predict:negative  (b) Predict:positive  (c) Predict:positive  (d) Predict:negative

Figure 5: Importance visualization for the UAT case study. (a): the original input by the original prefix-tuning; (b): the UAT-attacked input by the original prefix-tuning. Compared with (a), the trigger tokens in (b) attract major importance. (c) and (d): the UAT-attacked input by our robust prefix-tuning with (c) $N = 24$ and (d) $N = 3$. For $N = 3$, the LM is steered to ignore the distraction of the trigger tokens and assign high importance to "filler" at the time step of token ".".

LM is steered to *ignore the distraction* from the trigger and assign the highest importance to the most essential token as the baseline does with clean inputs. The findings are also supported by quantitative studies. By designing metrics of corpus-level Degree of Distraction (cDoD) and Recognition of the Essential token (cRoE), we find that the scores of our framework with $N = 3$ are better than those of other methods with statistical significance ($p < 0.001$), which also explains the results in Figure 3. We attach more visualization and details of the quantitative studies in Appendices F.2.2 and F.2.3.

## 6 Conclusion and Future Work

In this work, we propose robust prefix-tuning for text classification that maintains only one extra amount of embedding parameters of prefix-tuning while achieving substantial robustness improvement. We conduct both quantitative and qualitative studies to analyze the behavior of our framework. We also interpret prefix-tuning and our framework from the optimal control perspective. Future work includes constructing better canonical manifolds and extending our framework to text generation.

## ETHICS STATEMENT

Large-scaled pretrained language models have been criticized in terms of bias and stereotypes (Zhao et al., 2019; Bender et al., 2021; Field et al., 2021), and can be steered to generate malicious outputs with universal adversarial triggers (Wallace et al., 2019). The lack of robustness leads to more uncertain behavior of LM, especially under text attacks. In our work, we contribute to improving the robustness of LM with the robust prefix-tuning framework. Figure 11 illustrates an example of LM fooled to predict a gender-biased sentence as positive. With our framework, the prediction of the LM is rectified. The huge resource consumption of training large-scaled LMs has also led to sustainability concerns (Strubell et al., 2019). As specified in Appendix B and Table 11, our framework has preserved the efficiency and modularity of prefix-tuning by requiring significantly less time than all baseline methods and keeping the pretrained models unmodified.

## REPRODUCIBILITY STATEMENT

For all of our experiments, we have listed the detailed settings in Appendices B and C.1. For the theory part, we have provided our detailed formulation as well as a brief introduction to the optimal control theory in Appendix G. We have also provided the statistics of the datasets used, as well as the time and storage taken by all methods in Tables 10 and 11 and Appendix B, respectively.

## ACKNOWLEDGEMENTS

This work was supported by the National Key R&D Program of China (No. 2018YFB1005103), National Natural Science Foundation of China (No.61925601), and Huawei Noah's Ark Lab. We thank all anonymous reviewers for their valuable comments and suggestions on this work.

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

# A    COMPARISON WITH REGULAR FINETUNING

In this section, we provide comparison between our work with both standard and adversarial regular finetuning baselines on SST-2. For regular finetuning experiments, we use the 345M GPT2-medium LM, same as the LM adopted in the prefix-tuning experiments. We trained for 9 epochs for standard regular finetuning and 3 epochs for adversarial regular finetuning (earlystopping). The initial learning rate is also set as 5e-5 for both standard and adversarial regular finetuning. The settings of perturbation range $\epsilon$, step size $\alpha$ and iteration of the inner maximization $M$ is the same as the those adopted in adversarial prefix-tuning experiments. The results are listed in Table 6.

Table 6: Performance of our robust prefix-tuning with regular finetuning baselines for comparison.

| Method | #Epochs | Clean | PWWS | VIPER | SCPN | BUG | UAT |
|---|---|---|---|---|---|---|---|
| std. regular finetuning | 9 | 93.68 | 24.77 | 4.61 | 30.15 | 17.74 | 23.89 |
| std. prefix-tuning | 100 | 92.48 | 16.64 | 1.92 | 31.58 | 8.84 | 5.05 |
| + our framework | - | **92.59** | **50.36** | **44.65** | **58.54** | **46.68** | **85.72** |
| adv. regular finetuning | 3 | 93.63 | 48.38 | 33.44 | 48.27 | 44.59 | 14.17 |
| adv. prefix-tuning | 20 | 89.51 | 32.02 | 17.35 | 43.33 | 27.38 | 8.57 |
| + our framework | - | **89.57** | **53.93** | **48.38** | **57.88** | **49.70** | **73.97** |
| adv. prefix-tuning | 100 | 93.74 | 30.53 | 8.18 | 33.11 | 27.51 | 8.95 |
| + our framework | - | **93.79** | **57.55** | **43.60** | **60.68** | **57.17** | **91.05** |

Due to time limit, we have only conducted regular finetuning experiments on SST-2 with one adversarial training baseline (Miyato et al., 2017) for comparison. The aim of the comparison, however, is **neither** to set a new SOTA **nor** to demonstrate that our robust prefix-tuning framework has beaten the robust finetuning techniques. On the contrary, we aim to demonstrate several special properties in the scenario of prefix-tuning, as well as challenges and opportunities in robustness:

- Difficulty in optimization. It can be seen that prefix-tuning requires more epochs to converge. A potential explanation is that the loss landscape of the prefix parameters is highly non-convex, as the downstream task should be learned with far fewer amount of free parameters. The difficulty in optimization might have also brought more challenges on robustness to prefix-tuning. According to Table 6, all prefix-tuning baselines underperform the corresponded regular finetuning approaches against almost all types of attacks.

- Trade-off between space and time. It is worthy to be mentioned that for each prefix-tuning experiment, the saved prefix parameters consume disk storage of only 2MB. In contrast, each regular finetuning experiment takes 725MB disk storage. However, prefix-tuning also takes more epochs (and thus longer time) to converge. This brings even more challenges in robustifying prefix-tuning, as an ideal solution should be **neither** harmful to the benefits of prefix-tuning (lightweightness, modularity, not modifying the LM parameters) **nor** too slow for fear that the time complexity of prefix-tuning further deteriorates. Our framework serves as a possible solution that keeps the strengths without weakening the weakness.

- Lack of robustness against UAT. According to Table 6, both adversarial regular finetuning and adversarial prefix-tuning fail to defend against UAT. A possible explanation is that the adversaries induced during adversarial training are more similar to in-sentence attacks, yet few inductive bias has been introduced against UAT. Existing defense approaches against UAT maintain additional adversary detectors. For example, DARCY (Le et al., 2021) searches for the potential triggers first, and then retrain a classifier using the exploited triggers as a UAT adversary detector; T-Miner (Azizi et al., 2021) leverages Seq2Seq model to probe the hidden representation of the suspicious classifier into a synthetic text sequence that is likely to contain adversarial triggers. In addition, neither of the methods have been tested on large-scale pretrained LM. To the best of our knowledge, our framework is the first solution to defend against UAT by tuning an additional prefix based on prefix-tuning for pretrained LM. Positioned before the adversarial triggers, the robust prefix regulates the activation at the output position so that "the distraction can be ignored".

## B  DETAILS OF EXPERIMENTAL SETTINGS

We use the SST-2, AG's News, and SNLI benchmark for our text classification experiments. We use the 345M GPT2-medium (725MB storage) as the large-scale pretrained LM and implement prefix-tuning with prefix length of 10. We initialize the prefix token embeddings with parameters from the LM word embedding matrix to ease the optimization. We train 100 epochs for SST-2 and 25 epochs for AG's News and SNLI. We use the AdamW optimizer (Loshchilov & Hutter, 2019) provided by the HuggingFace Transformers library (Wolf et al., 2020) to optimize the prefix with initial learning rate as 5e-5 in all experiments. Other settings of prefix-tuning follows Li & Liang (2021). During inference in our framework, unless specified (Section 4.5 and Appendix E), we use an adaptive test batch size with dynamic padding to make full use of GPU memory. The total number of tokens in a test batch is determined by the setting of the ratio of GPU memory (0.12 for 24GB NVIDIA-3090 GPUs) allocated for loading data. In contrast, the number of sentences in a test batch is fixed in experiments in Section 4.5 and Appendix E.

The size of obtained prefix vectors are 1.9MB for all tasks. We also save the projection matrices of the layerwise canonical manifolds, but only the matrices of the bottom $N = 3$ canonical manifolds need to be saved as we tune the bottom $N = 3$ layers in our robust prefix-tuning. The saved matrices take 12MB storage and can be used for all types of text attacks. We use the Adam optimizer (Kingma & Ba, 2015) to tune the additional prefix $P'_\psi[i, :]$. The initial learning rate and the number of steps (5 or 10) for its optimization are tuned on the development set.

We use the following five text attack approaches in our experiments:

- PWWS (Ren et al., 2019), a score-based word-level text attack that fools a text classifier by applying greedy word substitution on inputs.
- VIPER (Eger et al., 2019), a blind character-level text attack that applies visually similar character substitution on inputs.
- SCPN (Iyyer et al., 2018), a blind sentence-level text attack that paraphrases inputs.
- TextBugger (Li et al., 2019), a gradient-based word- and character-level attack that applies greedy word substitution and character manipulation. We refer to it as "BUG" for short.
- UAT (Wallace et al., 2019), a gradient-based word-level text attack that fools a text classifier by prepending the same adversarial tokens to all test inputs.

We use the OpenAttack toolkit (Zeng et al., 2021) for the in-sentence attacks (PWWS, VIPER, SCPN, TextBugger) and implement UAT by ourselves. For the in-sentence attacks, we use the default hyperparameter settings provided by OpenAttack. Following prior arts (Ren et al., 2019; Jia et al., 2019; Dong et al., 2021), we do not attack the premise on SNLI. For UAT, following Wallace et al. (2019), we search for a 3-token trigger for inputs of each class in the test set. We set beam size as 5 and iterate over the test set for 5 epochs. We use NVIDIA-3090 GPUs for all of our experiments.

## C  ADVERSARIAL TRAINING

### C.1  ADVERSARIAL TRAINING BASELINES

We start with adversarial training for text classification that restricts perturbation within $\ell_2$ balls of word or sentence level (Miyato et al., 2017). The objective is defined as

$$\min_\theta \mathbb{E}_{(x,y)\sim\mathcal{D}_{tr}} \left[ \max_{\|r\|_2\leq\epsilon} \mathcal{L}(y|\text{emb}(x) + r; \theta) \right], \tag{9}$$

where $r$ is the perturbation injected to the word embeddings of $x$. The inner maximization of $r$ can be implemented by projected gradient descent (Madry et al., 2018) with $M$ iterative updates:

$$r \leftarrow \text{Proj}_{\{\|r\|_2\leq\epsilon\}} \left[ r + \alpha \frac{\nabla_{\text{emb}(x)}\mathcal{L}(y|\text{emb}(x) + r; \theta)}{\|\nabla_{\text{emb}(x)}\mathcal{L}(y|\text{emb}(x) + r; \theta)\|_2} \right]. \tag{10}$$

In all of our adversarial prefix-tuning experiments, we set $\epsilon = 5$, $\alpha = 1.25$, and $M = 10$.

For texts, the $\ell_2$ ball can be defined for both word level and sentence level. For the word level, the $\ell_2$ balls are constructed for each token in the context part of the input by offsetting each word vector

Table 7: Results of different baselines with our framework applied. This table covers the results of adversarial prefix-tuning with both word- and sentence-level $\ell_2$ balls and serves as the supplementary for Table 1. Our framework consistently improves robustness of all baselines on all benchmarks.

| Benchmark | Method | Clean | PWWS | VIPER | SCPN | BUG | UAT |
|---|---|---|---|---|---|---|---|
| SST-2 | std. prefix-tuning | 92.48 | 16.64 | 1.92 | 31.58 | 8.84 | 5.05 |
| | + our framework | **92.59** | **50.36** | **44.65** | **58.54** | **46.68** | **85.72** |
| | word-level adv. | 93.74 | 30.53 | 8.18 | 33.11 | 27.51 | 8.95 |
| | + our framework | **93.79** | **57.55** | **43.60** | **60.68** | **57.17** | **91.05** |
| | sent-level adv. | **93.57** | 30.64 | 7.25 | 35.42 | 25.04 | 4.88 |
| | + our framework | 93.52 | **53.21** | **39.65** | **54.91** | **50.69** | **88.25** |
| AG News | std. prefix-tuning | 86.42 | 43.00 | 24.55 | 43.20 | 43.22 | 52.20 |
| | + our framework | **86.50** | **53.91** | **24.93** | **48.38** | **51.79** | **76.47** |
| | word-level adv. | 89.46 | 50.91 | 26.30 | 45.25 | 47.11 | 59.74 |
| | + our framework | **90.26** | **56.45** | **31.25** | **48.03** | **54.66** | **87.26** |
| | sent-level adv. | 90.18 | 49.97 | 24.87 | 44.54 | 46.41 | 38.11 |
| | + our framework | **90.37** | **57.75** | **26.50** | **48.43** | **54.39** | **80.63** |
| SNLI | std. prefix-tuning | 72.48 | 25.51 | 29.69 | 42.46 | 32.88 | 30.20 |
| | + our framework | **72.74** | **34.68** | **33.64** | **43.37** | **36.59** | **71.03** |
| | word-level adv. | 77.22 | 27.43 | 28.58 | 46.83 | 34.94 | 35.76 |
| | + our framework | **77.65** | **33.88** | **33.98** | **46.92** | **38.21** | **76.15** |
| | sent-level adv. | 78.99 | 25.75 | 28.26 | 44.54 | 31.94 | 41.77 |
| | + our framework | **79.56** | **30.91** | **34.16** | **44.66** | **36.41** | **77.47** |

within the range of radius of $\epsilon$. For the sentence level, the single $\ell_2$ ball is constructed for the entire context part of the input by flattening the word embedding vectors and offsetting the flattened vector within the range of radius $\epsilon$. The word-level and the sentence-level prefix-tuning take similar training time. As a consequence, adversarial prefix-tuning with both levels of $\ell_2$ balls are far slower than standard prefix-tuning. The adversarial training approach used in Table 1 is of word-level $\ell_2$ balls. The complete results of our approach applied with both word-level and sentence-level adversarial prefix tuning, as well as standard prefix-tuning, are shown in Table 7. It can be seen that our robust prefix-tuning framework substantially improves robustness of different baseline methods.

We have also experimented with other types of adversarial training for prefix-tuning, such as adding the KL-divergence term for regularization (Miyato et al., 2019; Zhang et al., 2019). The objective is

$$\min_{\theta} \mathbb{E}_{(x,y)\sim\mathcal{D}_{tr}} \left[ \mathcal{L}(y|x;\theta) + \max_{\|r\|_2 \leq \epsilon} \beta\mathrm{KL}\left[\mathcal{L}(y|\mathrm{emb}(x);\theta) \parallel \mathcal{L}(y|\mathrm{emb}(x)+r;\theta)\right] \right]. \quad (11)$$

We set $\beta = 1$ and $\beta = 4$ and train for 20 epochs for different attempts of KL-divergence regularized adversarial prefix-tuning with Eq. (11). The training loss and the clean accuracy of the development set on the SST-2 benchmark is shown in Figure 6. In this figure, we also visualize the word-level adversarial prefix-tuning method shown in Figure 2 for comparison. The KL-divergence regularized adversarial prefix-tuning of both settings start with very large training loss. For $\beta = 1$, the clean accuracy on the development set remains 0 for the first 15 epochs. The accuracy raises to around 27% at the 17-th epoch, but lacks stability (back to 0 again at the 19-th epoch). For $\beta = 4$, the clean accuracy remains 0 for all 20 epochs. In comparison, the word-level adversarial prefix-tuning used in Section 4.2 achieves low training loss and high clean accuracy on the development set. The training loss and the accuracy on the development set have suggested the difficulty in optimizing the prefix embedding, which might be attributed to its rather small amount of parameters compared with the whole language model. While there are other variants of adversarial training approaches for text classification, in this work, we did not use them for our baselines as most of them (Zhu et al., 2020; Jiang et al., 2020a; Liu et al., 2020; Dong et al., 2021) are based on the KL-divergence regularizer, and also require excessively long training time. It is also of future interest on how to empower our robust prefix-tuning framework with faster and better adversarial prefix-tuning approaches.

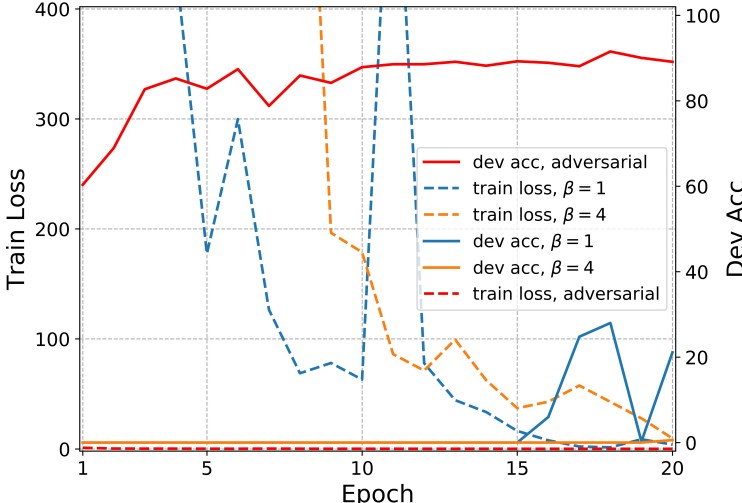

Figure 6: Training loss and clean accuracy on the dev set of SST-2 with KL-divergence regularized adversarial prefix-tuning as well as the word-level adversarial prefix-tuning used in Section 4.2.

## C.2 ADVERSARIAL TRAINING WITH EARLYSTOPPING

We apply earlystopping to both word-level and sentence-level adversarial prefix-tuning approaches on the SST-2 benchmark at 20 epochs. The results are shown in Table 8. According to the results, the robustness results against different types of in-sentence attacks have been slightly improved. The robustness against UAT, in contrast, slightly drops. In terms of clean accuracy, the earlystopped adversarial prefix-tuning is even surpassed by the standard prefix-tuning baseline. Our robust prefix-tuning framework still works on the prefixes obtained by the earlystopped adversarial prefix-tuning approaches, as the robustness results are consistently improved.

Table 8: Results of earlystopping of adv. prefix-tuning at 20 epochs with our framework on SST-2.

| Method | Clean | PWWS | VIPER | SCPN | BUG | UAT |
|---|---|---|---|---|---|---|
| std. prefix-tuning | 92.48 | 16.64 | 1.92 | 31.58 | 8.84 | 5.05 |
| + our framework | **92.59** | **50.36** | **44.65** | **58.54** | **46.68** | **85.72** |
| 100-epoch word-level | **93.57** | 30.64 | 7.25 | 35.42 | 25.04 | 4.88 |
| + our framework | 93.52 | **53.21** | **39.65** | **54.91** | **50.69** | **88.25** |
| 100-epoch sent-level | 93.74 | 30.53 | 8.18 | 33.11 | 27.51 | 8.95 |
| + our framework | **93.79** | **57.55** | **43.60** | **60.68** | **57.17** | **91.05** |
| 20-epoch word-level | **90.88** | 34.71 | 22.41 | 39.37 | 29.00 | 6.97 |
| + our framework | **90.88** | **51.18** | **46.46** | **58.26** | **50.74** | **84.95** |
| 20-epoch sent-level | 89.51 | 32.02 | 17.35 | 43.33 | 27.38 | 8.57 |
| + our framework | **89.57** | **53.93** | **48.38** | **57.88** | **49.70** | **73.97** |

## D    ADVERSARIAL DATA AUGMENTATION

In this section, we report additional experimental results of adversarial data augmentation using VIPER or SCPN to generate adversarial data for augmentation during training. We also set the milestones of 20, 50, and 100 epochs for training. According to Table 9, the robustness against the attack of the same type used for adversarial data augmentation during training has gained significant improvements compared with the standard prefix-tuning baseline. Robustness against other types of attacks, however, fails to improve with the process of training. For our approach, we use the correctly classified data from both the clean training set and the training data perturbed by the specific type of attack used for augmentation to construct the canonical manifolds. The experimental results indicate that our robust prefix-tuning framework still improves the robustness against all types of text attacks over all types of baselines (except for the VIPER attack in VIPER adversarial data augmentation that incurs slight drop at the 20-epoch and 50-epoch milestones).

Table 9: Results of VIPER and SCPN data augmentation baselines as well as our methods on SST-2.

| Method | Clean | PWWS | VIPER | SCPN | BUG | UAT |
|---|---|---|---|---|---|---|
| std. prefix-tuning | 92.48 | 16.64 | 1.92 | 31.58 | 8.84 | 5.05 |
| + our framework | **92.59** | **50.36** | **44.65** | **58.54** | **46.68** | **85.72** |
| 20-epoch VIPER aug. | 82.59 | 38.93 | **53.32** | 50.85 | 37.51 | 10.49 |
| + our framework | **83.09** | **50.03** | 51.89 | **54.09** | **50.91** | **59.75** |
| 50-epoch VIPER aug. | **90.06** | 36.02 | **63.64** | 45.47 | 30.97 | 21.03 |
| + our framework | 89.73 | **51.89** | 63.21 | **58.15** | **47.83** | **81.49** |
| 100-epoch VIPER aug. | **93.63** | 25.54 | 52.22 | 36.19 | 21.75 | 5.22 |
| + our framework | 92.97 | **51.18** | **52.61** | **59.75** | **49.53** | **84.57** |
| 20-epoch SCPN aug. | 72.27 | 41.35 | 40.14 | 54.64 | 42.17 | 16.20 |
| + our framework | **72.38** | **50.74** | **49.97** | **54.75** | **50.80** | **62.11** |
| 50-epoch SCPN aug. | **87.64** | 34.32 | 43.33 | 62.93 | 35.04 | 10.38 |
| + our framework | **87.64** | **48.82** | **48.76** | **63.98** | **49.82** | **82.43** |
| 100-epoch SCPN aug. | 89.13 | 28.45 | 39.32 | 59.91 | 23.39 | 2.69 |
| + our framework | **89.24** | **51.67** | **48.60** | **62.11** | **48.33** | **79.35** |

We attach the clock time that each method takes on each benchmark for comparison in Table 11. The time differs for each benchmark because of different sizes (Table 10). The training-phase adversarial prefix-tuning takes far greater time than standard prefix-tuning. The adversarial data augmentation methods during training phase take the longest time due to the computationally-challenging discrete search/optimization of text attacks. For our framework, steps 1 and 2 that construct the canonical manifolds are viewed as preprocessing since it can be used against all attacks during inference. As $\|S_C\| >> d$ (mentioned in Section 3), we can further speed up step 1 by random sampling in $S_C$. The additional prefix $P'_\psi[i, :]$ is tuned on the fly during test phase in step 3 of our framework, which slackens the standard testing but is negligible compared with the training-phase baseline methods. In conclusion, our framework has preserved the efficiency and modularity of prefix-tuning.

Table 10: Dataset statistics for each benchmark. We have also included the number of classes in each benchmark and the accuracy of random classifier in theory for better understanding.

| Benchmark | #Classes | Train | Dev | Test | Random Acc(%) |
|---|---|---|---|---|---|
| SST-2 | 2 | 6,920 | 872 | 1,821 | 50.00 |
| AG's News | 4 | 115,000 | 5,000 | 7,600 | 25.00 |
| SNLI | 3 | 550,152 | 10,000 | 10,000 | 33.33 |

Table 11: Time used by different methods for all benchmarks. Compared with the time-consuming training-phase baseline methods, our test-phase robust prefix-tuning is significantly faster.

| Phase | Method | Time (GPU min) | | |
|---|---|---|---|---|
| | | SST-2 | AG's News | SNLI |
| Train | std. prefix-tuning | 3.7×100 | 19.7×25 | 55.4×25 |
| | adv. prefix-tuning | 17.7×100 | 223.0×25 | 641.1×25 |
| | adv. aug. PWWS | 82.5×100 | - | - |
| | adv. aug. VIPER | 47.3×100 | - | - |
| | adv. aug. SCPN | 285.3×100 | - | - |
| Preprocess | ours: steps 1 and 2 | 1.0 | 69.9 | 721.5 |
| Test | std. testing | 0.3 | 1.2 | 0.7 |
| | ours: step 3 (adaptive) | 1.0 | 6.5 | 6.5 |

# E  MORE REALISTIC EVALUATION SETTINGS

## E.1  PERFORMANCE WITH SMALL TEST BATCH SIZES

In this section, we continue to study why our methods with test batch size of 2 or 4 achieve stronger robustness against UAT (shown in Table 3). We note that the $j$-th layer activation at the output position $H_T^{(j)}$ is normalized by rows before being projected to the canonical manifold $\mathcal{M}^{(j)}$ when the test batch size is larger than $> 1$. According to Eq. (8) and the description in Section 3, when the size of a test batch is larger than 1, namely, when

$$H_T^{(j)} = \left[ h_{o,t}^{(j)} \right] \in \mathbb{R}^{|S_T| \times d} \tag{12}$$

has more than 1 row ($|S_T| > 1$), the objective for tuning $P'_\psi$ is

$$\min_\psi \sum_{j=0}^{N-1} L\left( \psi^{(j)}, Q^{(j)} \right) = \sum_{j=0}^{N-1} \left\| \left( H_T^{(j)} - \mathbf{1}\mathbf{1}^{\mathrm{T}} H_T^{(j)}/|S_T| \right) \left( I - Q^{(j)} \right) \right\|_2. \tag{13}$$

The normalization is performed to alleviate the randomness among the samples in the test batch. It is not applicable when $|S_T| = 1$, as $H_T^{(j)} - \mathbf{1}\mathbf{1}^{\mathrm{T}} H_T^{(j)}/|S_T|$ would be $\mathbf{0}$. Therefore, for test batch size $= 1$, the loss for tuning $P'_\phi$ is Eq. (8) without normalization before projection.

We conduct experiments with test batch sizes of 2 and 4 without normalizing the test batch before projection during inference for comparison. According to the results in Table 12, for test batch sizes of 2 and 4, robustness results against UAT have degenerated without the normalization, but they are still better than that with test batch size of 1 as mini-batch optimization may achieve better results than the fully online setting. The robust performance of our framework against other in-sentence attacks remains similar, with or without normalizing.

Table 12: Performance of our method with std. prefix-tuning with various test batch sizes on SST-2. For test batch sizes of 2 and 4, we compare the robustness performance of our framework with and without normalization before projection during inference.

| Method | Clean | PWWS | VIPER | SCPN | BUG | UAT |
|---|---|---|---|---|---|---|
| std. prefix-tuning | 92.48 | 16.64 | 1.92 | 31.58 | 8.84 | 5.05 |
| ours, bsz adaptive | 92.59 | 50.36 | 44.65 | 58.54 | 46.68 | 85.72 |
| ours, bsz = 1 (unnormalized) | **92.64** | **51.46** | 43.99 | 58.32 | 46.73 | 73.09 |
| ours, bsz = 2 (normalized) | 92.48 | 49.92 | 45.74 | **59.14** | 48.93 | 84.73 |
| ours, bsz = 4 (normalized) | **92.64** | 50.36 | **45.85** | 58.54 | **49.31** | **86.66** |
| ours, bsz = 2, unnormalized | 92.53 | 51.13 | 42.89 | 57.99 | 48.00 | 78.20 |
| ours, bsz = 4, unnormalized | 92.59 | 51.84 | 43.44 | 58.59 | 47.89 | 80.23 |

The results lead to a temporary conclusion that normalization before projection during inference can be important when using our framework to defend against UAT. However, the performance gap between the unnormalized test batch size $= 1$ setting and the normalized test batch size $> 1$ setting is much smaller on other benchmarks and other prefix-tuning baselines. We set test batch size as 1 and evaluate the performance of our framework with both standard and adversarial prefix-tuning on both SST-2 and AG's News benchmarks. The results in Table 13 show that the robustness gap against UAT between the unnormalized test batch size $= 1$ setting and the normalized test batch size $> 1$ setting, though still exists, is relatively small compared with the results in Table 12.

As a result, for conventional evaluation where accuracy under specific attack is reported, we will use test batch size larger than 1 (2 or 4 should be satisfactory), as normalization before projection for additional tuning during inference can be beneficial. In contrast, we will set test batch size as 1 in more realistic settings, as it is able to cope with real-time or mixed test data.

Our another attempt for improvement is to set the normalization before projection during inference with pre-computed vectors beforehand. Currently the normalization $\mathbf{1}\mathbf{1}^{\mathrm{T}} H_T^{(j)}/|S_T|$ in Eq. (13)

Table 13: Performance of our framework with both standard and adversarial prefix-tuning on both SST-2 and AG's News benchmarks, with test batch size of 1 or adaptive.

| Benchmark | Method | Clean | PWWS | VIPER | SCPN | BUG | UAT |
|-----------|--------|-------|------|-------|------|-----|-----|
| SST-2 | std. prefix-tuning | 92.48 | 16.64 | 1.92 | 31.58 | 8.84 | 5.05 |
| | + ours, bsz adaptive | 92.59 | 50.36 | **44.65** | **58.54** | **46.68** | **85.72** |
| | + ours, bsz = 1 | **92.64** | **51.46** | 43.99 | 58.32 | 46.73 | 73.09 |
| | adv. prefix-tuning | 93.57 | 30.64 | 7.25 | 35.42 | 25.04 | 4.88 |
| | + ours, bsz adaptive | 93.79 | **57.55** | 43.60 | **60.68** | **57.17** | **91.87** |
| | + ours, bsz = 1 | **94.12** | 56.67 | **46.13** | 60.30 | 55.19 | 87.37 |
| AG's News | std. prefix-tuning | 86.42 | 43.00 | 24.55 | 43.20 | 43.22 | 52.20 |
| | + ours, bsz adaptive | 86.50 | **53.91** | 24.93 | **48.38** | **51.79** | **76.47** |
| | + ours, bsz = 1 | **86.75** | 46.47 | **35.82** | 45.61 | 49.00 | 73.08 |
| | adv. prefix-tuning | 89.46 | 50.91 | 26.30 | 45.25 | 47.11 | 59.74 |
| | + ours, bsz adaptive | **90.26** | **56.45** | 31.25 | 48.03 | **54.66** | **87.26** |
| | + ours, bsz = 1 | 89.79 | 55.72 | **34.33** | **49.43** | 53.75 | 87.16 |

depends on $H_T^{(j)}$, with limitation in the setting of test batch size = 1 and concern in reproducibility. In our additional experiments, we pre-compute the normalization of the $j$-th layer activation during inference using $S_C$, the set of correctly classified training examples. Consequently, we can replace $\mathbf{1}\mathbf{1}^{\mathrm{T}} H_T^{(j)}/|S_T|$ in Eq. (13) with $\mathbf{1}\mathbf{1}^{\mathrm{T}} H_C^{(j)}/|S_C|$, which can be obtained in the second step in Section 3. In this way, the normalization is static for different batches and applicable to test batch with the size of 1. The results are shown in Table 14.

Table 14: Performance of our method with std. prefix-tuning with static or dynamic normalization before projection during inference on SST-2.

| Method | Clean | PWWS | VIPER | SCPN | BUG | UAT |
|--------|-------|------|-------|------|-----|-----|
| std. prefix-tuning | 92.48 | 16.64 | 1.92 | 31.58 | 8.84 | 5.05 |
| ours, bsz adaptive | 92.59 | 50.36 | 44.65 | 58.54 | 46.68 | 85.72 |
| ours, bsz = 1 (unnormalized) | **92.64** | **51.46** | 43.99 | 58.32 | 46.73 | 73.09 |
| ours, bsz = 2 (dynamic norm.) | 92.48 | 49.92 | 45.74 | **59.14** | 48.93 | 84.73 |
| ours, bsz = 4 (dynamic norm.) | **92.64** | 50.36 | **45.85** | 58.54 | **49.31** | 86.66 |
| ours, bsz = 1 static norm. | 92.53 | 49.48 | 40.25 | 58.43 | 48.98 | **87.92** |
| ours, bsz = 2, static norm. | 92.37 | 50.58 | 41.57 | 58.48 | 47.17 | 87.70 |
| ours, bsz = 4, static norm. | 92.48 | 49.70 | 40.69 | 58.15 | 48.22 | 87.64 |

On the one hand, it can be seen that our framework with static normalization outperforms the ones with dynamic normalization under UAT. The improvement is especially substantial for the setting with test batch size = 1, indicating that while the dynamic normalization $\mathbf{1}\mathbf{1}^{\mathrm{T}} H_T^{(j)}/|S_T|$ is not applicable for test batch size = 1, the static normalization $\mathbf{1}\mathbf{1}^{\mathrm{T}} H_C^{(j)}/|S_C|$ can be helpful in this setting. On the other hand, for in-sentence attacks, the robustness results are comparable between static and dynamic normalization, though there is a performance drop under VIPER. Due to time limit, we have only experimented on SST-2 with standard prefix-tuning as the baseline. We will conduct more experiments to further evaluate static normalization with other baselines on different benchmarks. In the following experiments under more realistic threat models, we adopt the setting of the test batch size = 1 without normalization before projection.

## E.2 PERFORMANCE UNDER MIXED TEST DATA

In this section, we provide more experimental results of our framework under mixed test data. We combine the clean test set with the test set under one attack, or perturb the test set with two different attacks separately and combine the two sets as mixed test data. The mixed test data can be from a more realistic scenario. To cope with it, we set the test batch size as 1 in order to avoid the situation where both perturbed and unperturbed test sentences or perturbed sentences under different attacks exist in a test batch. Besides, by setting the test batch size as 1 we can deal with real-time data, which is also a practical approach. We use adversarial prefix-tuning as the baseline to obtain a strong prefix $P_\theta[i,:]$. While we are not able to tune the robust prefix under specific attack in this setting, we find it effective to tune the initial learning rate and the number of steps (5 or 10) for our inference-phase robust prefix on the UAT-attacked development set and fix them for every incoming test sentence. The results are listed in Table 15.

Table 15: Performance of our framework under mixed test data on SST-2 and AG's News. The '+' denotes combination of test set clean or under attack, and 'C' is short for "Clean", 'B' for "BUG", etc. The test batch size is set as 1 for all experiments with our framework. We also provide the averaged results with the robust prefix in our framework separately tuned on the two test sets as an upper bound for comparison.

| Benchmark | Method | C + B | C + P | V + B | V + P | S + B | S + P | U + B | U + P |
|-----------|--------|-------|-------|-------|-------|-------|-------|-------|-------|
| SST-2 | adv. PT | 60.65 | 62.27 | 17.76 | 20.43 | 29.57 | 31.85 | 18.12 | 20.81 |
| | + ours mixed | **69.71** | **71.11** | **46.02** | **47.23** | **55.63** | **56.92** | **70.04** | **71.67** |
| | ours, separate | 74.66 | 75.40 | 50.66 | 51.40 | 57.75 | 58.49 | 71.28 | 72.02 |
| AG's News | adv. PT | 52.38 | 53.41 | 26.54 | 27.14 | 34.86 | 35.36 | 40.97 | 42.16 |
| | + ours mixed | **71.36** | **71.39** | **42.84** | **43.99** | **49.04** | **50.12** | **69.88** | **71.30** |
| | ours, separate | 71.77 | 72.76 | 44.04 | 45.03 | 51.59 | 52.58 | 70.46 | 71.44 |

According to the results, the performance under mixed test data is consistently improved using our framework. We also provide the averaged results with the robust prefix in our framework separately tuned on the two test sets as an upper bound for comparison. It can be seen that small performance gap exists between the results under mixed data and the averaged results with separately tuning for specific test sets. A potential reason is that our framework can better adapt to the data under specific attack. To reduce the performance gap, we can construct more powerful canonical manifolds that capture richer data information of multiple granularity so that our framework also achieves optimal performance under mixed test data. We leave the exploration for future work.

# F    MODEL BEHAVIOR ANALYSIS

## F.1    IN-SENTENCE ATTACKS: AVERAGING THE ATTENTION

We provide the attention weights from the LM final layer over the entire $x$ = [context, question, [ANS]] with PWWS as the text attack from the development set of SST-2. The perturbed inputs are shown in Table 16, and the visualizations are shown in Figure 7, Figure 8, and Figure 9. Similar behavior of *averaging the attention* of the final layer in the LM can be also observed as the BUG attack visualized in Figure 4. **Caveat**: the observed behavior is *of the final layer only*, as attention is unreliable for indicating importance over input tokens (Jain & Wallace, 2019; Wiegreffe & Pinter, 2019; Abnar & Zuidema, 2020; Serrano & Smith, 2019).

Table 16: Original inputs from the development set of SST-2 with perturbation by PWWS.

| Type | Input (the context part) | Predict | Label |
|---|---|---|---|
| Original | a giggle a minute. | positive | positive |
| Perturbed | A giggle a arc minute. | negative | positive |
| Original | a marvel like none you've seen. | positive | positive |
| Perturbed | A wonder I none you've get. | negative | positive |
| Original | rarely has so much money delivered so little entertainment. | negative | negative |
| Perturbed | Rarely has so much money delivered so picayune entertainment. | positive | negative |

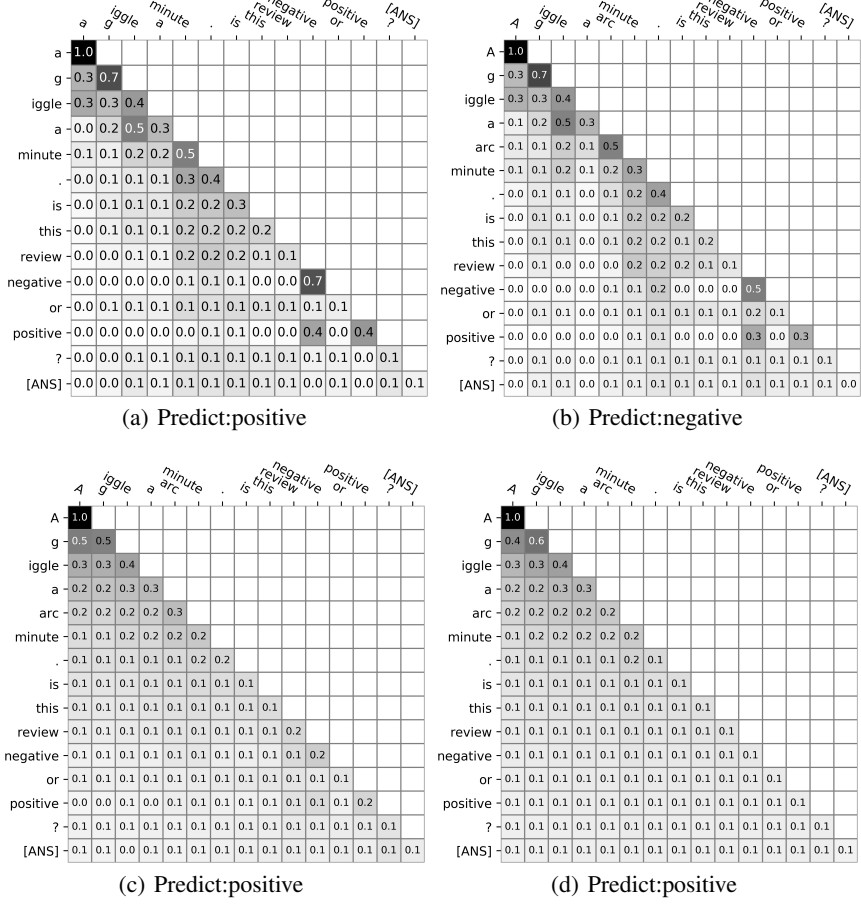

Figure 7: Visualized attention weights from the final layer in the LM for the first example in Table 16. (a): Original input with original prefix-tuning; (b): PWWS-perturbed input with original prefix-tuning; (c) and (d): PWWS-perturbed input with robust prefix-tuning of (c) $N = 24$ and (d) $N = 3$.

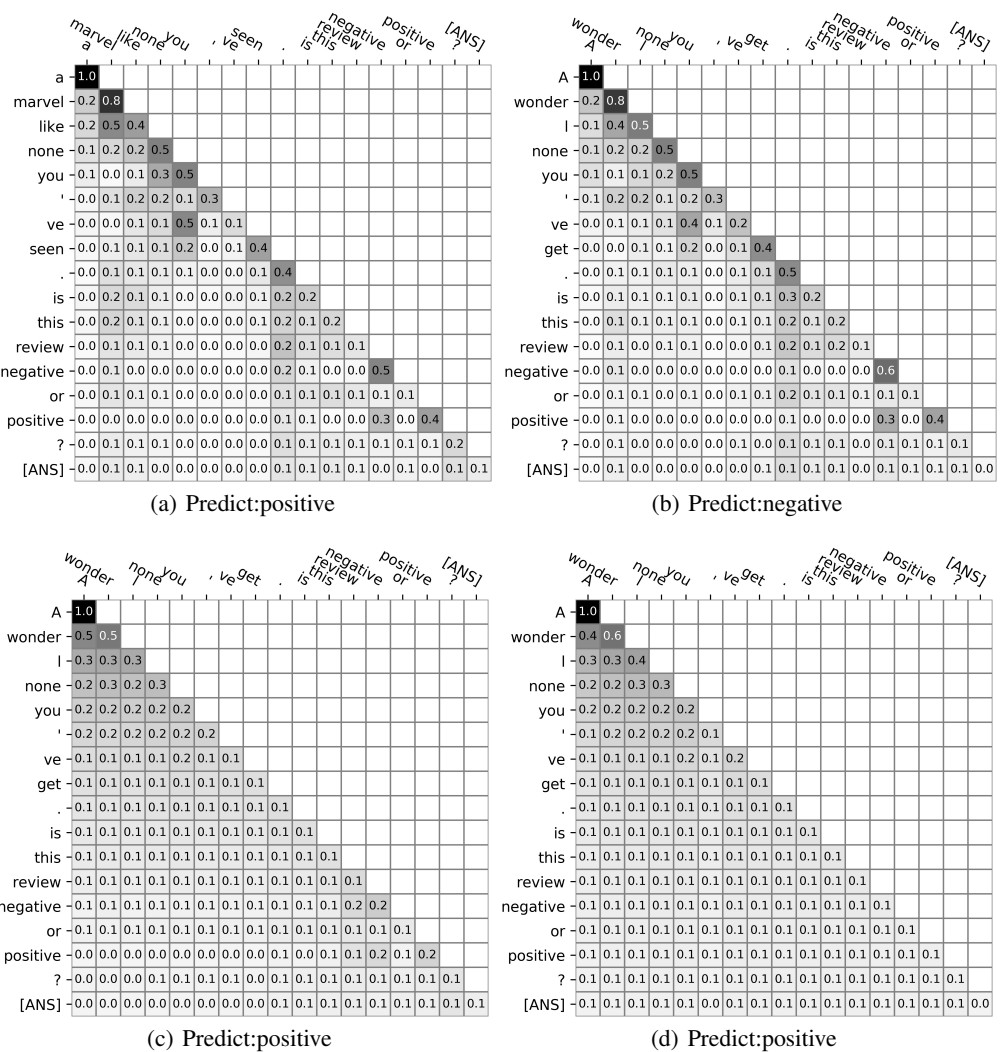

Figure 8: Visualized attention weights from the final layer in the LM for the second example in Table 16. (a): Original input with original prefix-tuning; (b): PWWS-perturbed input with original prefix-tuning; (c) and (d): PWWS-perturbed input with robust prefix-tuning of (c) $N = 24$ and (d) $N = 3$.

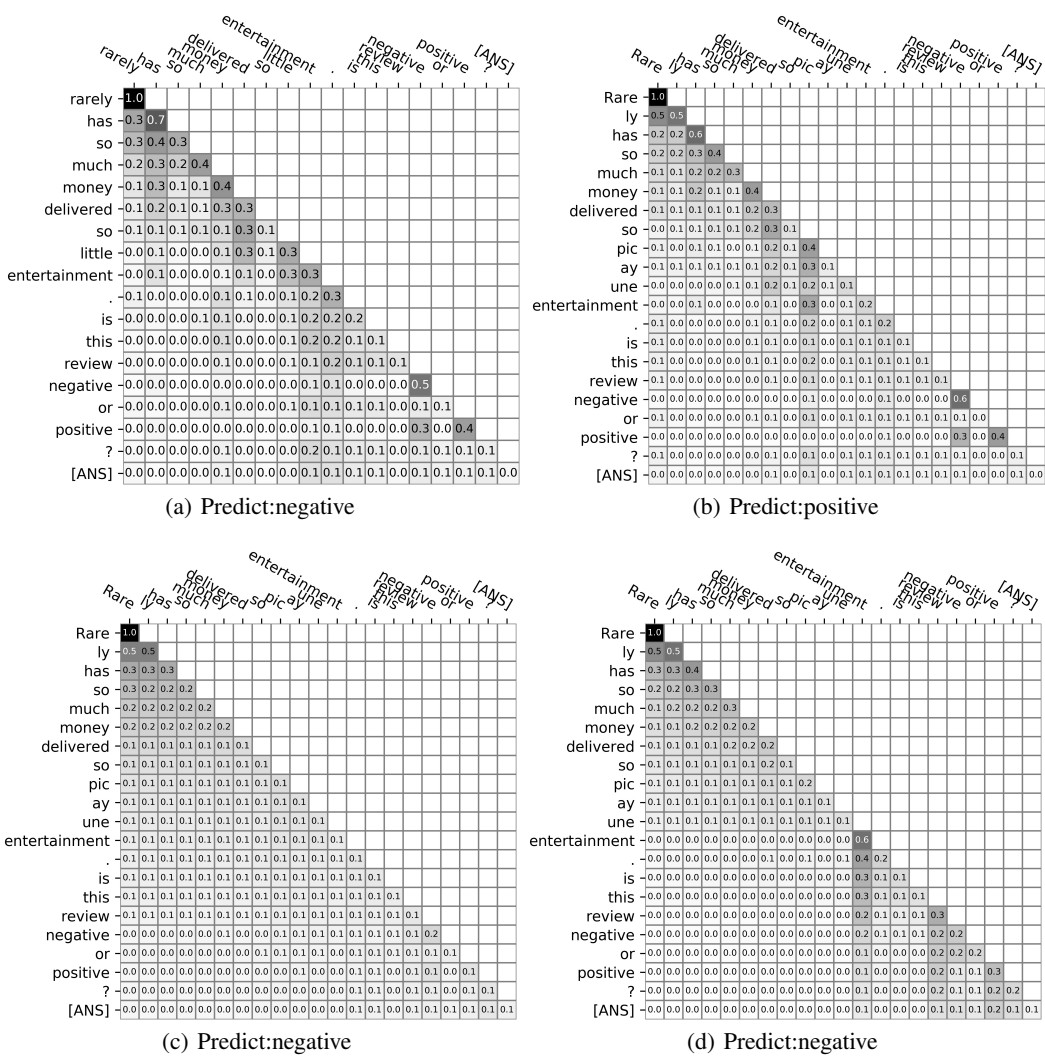

Figure 9: Visualized attention weights from the final layer in the LM for the third example in Table 16. (a): Original input with original prefix-tuning; (b): PWWS-perturbed input with original prefix-tuning; (c) and (d): PWWS-perturbed input with robust prefix-tuning of (c) $N = 24$ and (d) $N = 3$. In this example, robust prefix-tuning with $N = 24$ steers the final layer in the LM with the absolute behavior of *averaging the attention*, while that with $N = 3$ steers it to assign higher attention weights to the input vector at the position of the input token "entertainment" as well as that of each token in the question part. We use this visualization to show the slightly different behavior of our framework against in-sentence attacks when tuning different bottom $N$ layers.

## F.2 Universal Adversarial Triggers: Ignoring the Distraction

### F.2.1 The Alternative Proxy

In this section, we introduce the alternative proxy for the analysis of token importance as a replacement of attention. To interpret token importance, following Serrano & Smith (2019), we leverage an alternative gradient-based importance ranking proxy. As stated in Serrano & Smith (2019), while it does not provide a gold-standard indicator of importance value for each token to the model, the proxy is more reliable in reflecting the importance ordering for tokens in the sequence. In the proxy, the importance for each token is given by the multiplication of the gradient of the decision function with respect to each attention weight and the attention weight magnitude. As there are either 2, 3, or 4 classes in our text classification benchmarks, the decision function at the output position is given by

$$d_o \left( W \left( h_o^{(L)} \right) \right) = \frac{\exp \max_i \left[ W \left( h_o^{(L)} \right) \right]_i}{\sum_i \exp \left[ W \left( h_o^{(L)} \right) \right]_i}, \tag{14}$$

where $W$ in the LM transforms the top-layer output $h_o^{(L)}$ to a probability vector over the dictionary, and $i \in S_{\text{label}}$ traverses all the class label tokens in the vocabulary ($|S_{\text{label}}| = $ the number of classes in the benchmark). The decision function is proposed in Serrano & Smith (2019) to analyze hierarchical attention network (Yang et al., 2016). We further augment the decision function for time steps $< o$ to study the entire behavior of how the autoregressive LM assigns importance to each token visible at each time step. As introduced in Section 2, the autoregressive LM generates sequence $[x, \tilde{y}]$ with $[[\text{CLS}], x]$ as the input sequence for text classification tasks ($\tilde{y}$ is the generated label token as prediction). For each time step $< o$, the output of the LM is a token from $x$. As a result, we define the decision function at the $k$-th position as

$$d_k \left( W \left( h_k^{(L)} \right) \right) = \frac{\exp \left[ W \left( h_k^{(L)} \right) \right]_{x_k}}{\sum_{j \in \mathcal{V}} \exp \left[ W \left( h_k^{(L)} \right) \right]_j}, \tag{15}$$

with $x_{k-1}$ (or $[\text{CLS}]$ for $k = 0$) as input and $x_k$ as output. The complete decision function is then

$$d \left( W \left( h_0^{(L)} \right), \cdots, W \left( h_k^{(L)} \right), \cdots, W \left( h_o^{(L)} \right) \right) = \sum_{k=0}^{o} d_k \left( W \left( h_k^{(L)} \right) \right). \tag{16}$$

In this way, at each time step $i$, the gradient taken from the decision function (Eq. (16)) with respect to each attention weight $a_{ij}$ ($j \leq i$) in the final layer of the LM can be obtained with automatic differentiation in PyTorch (Paszke et al., 2019). We multiply the gradient with the attention weight magnitude as the value of token importance, which ranks a token with both high attention weight and high calculated gradient to be most important. Formally, for the attention weight on the $j$-th token at the $i$-th time step in the $h$-th head of the final layer (namely $a_{ij}^{(h)}$), its importance value is

$$I_{ij}^{(h)} = \frac{\partial d}{\partial a_{ij}^{(h)}} a_{ij}^{(h)}, \tag{17}$$

where $j \leq i$. The upper triangle part of the importance matrix is still masked, namely $I_{ij}^{(h)} = 0$ when $j > i$. We average the importance matrix over all heads in the final layer. Note that the averaged importance value can be negative, as each calculated gradient $\partial d / \partial a_{ij}^{(h)}$ in Eq. (17) can be negative. For better visualization, we further normalize all the token importance values at each time step $i$ by subtracting the minimum value so that the negative values are eliminated. We finally normalize all the (now positive) token importance values at each time step $i$ into a probability vector. Formally,

$$I_{ij}' = \frac{1}{H} \sum_h I_{ij}^{(h)}, \ \hat{I}_{ij} = I_{ij}' - \min_{0 \leq j \leq i} I_{ij}', \ I_{ij} = \frac{\hat{I}_{ij}}{\sum_{j=0}^{i} \hat{I}_{ij}}, \tag{18}$$

where $H$ is the number of heads in the final layer. The latter two equations transform $I'[i, :]$ to $I[i, :]$ as a probability vector. This is similar to an attention row, which is a better form for visualization. The importance matrix $I$ is plotted for all the case studies under UAT attack (Figure 5 and Appendix F.2.2). Also note that the transformation from $I'$ to $I$ is order-preserving for each row, and we will only use the importance ordering information in quantitative studies F.2.3.

F.2.2 QUALITATIVE ANALYSIS

We use UAT to attack inputs from the development set of SST-2 for visualization. The perturbed inputs are shown in Table 17, and importance visualizations are shown in Figures 10, 11, 12. Here we also plot the importance between each token pairs of the adversarial triggers. We can observe the behavior of *ignoring the distraction* for our robust prefix-tuning framework with $N = 3$.

Table 17: Original inputs from the development set of SST-2 with perturbation by UAT.

| Type | Input (the context part) | Predict | Label |
|---|---|---|---|
| Original | this is so bad. | negative | negative |
| Perturbed | lifts mates who this is so bad. | positive | negative |
| Original | it treats women like idiots. | negative | negative |
| Perturbed | lifts mates who it treats women like idiots. | positive | negative |
| Original | beautifully observed, miraculously unsentimental comedy-drama. | positive | positive |
| Perturbed | grades lame Invalid beautifully observed, miraculously unsentimental comedy-drama. | negative | positive |

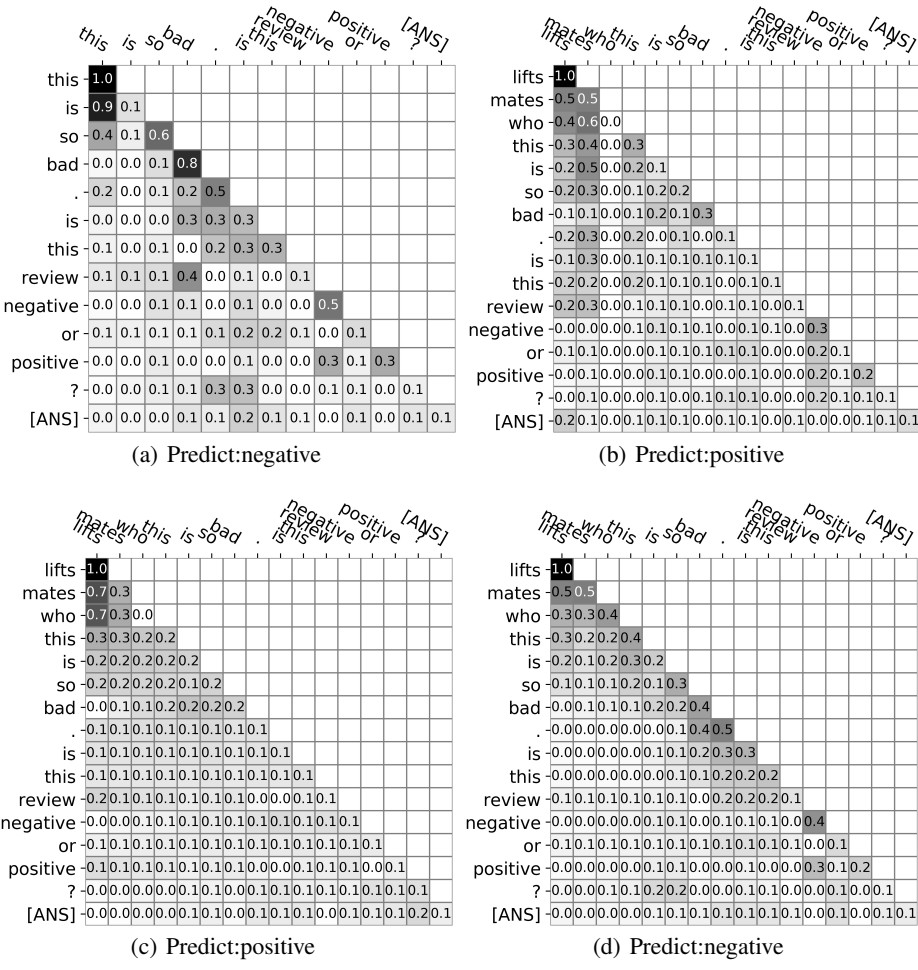

Figure 10: Visualized importance matrices for the first example in Table 17. (a): Original input with original prefix-tuning; (b): UAT-attacked input with original prefix-tuning; (c) and (d): UAT-attacked input with robust prefix-tuning of (c) $N = 24$ and (d) $N = 3$.

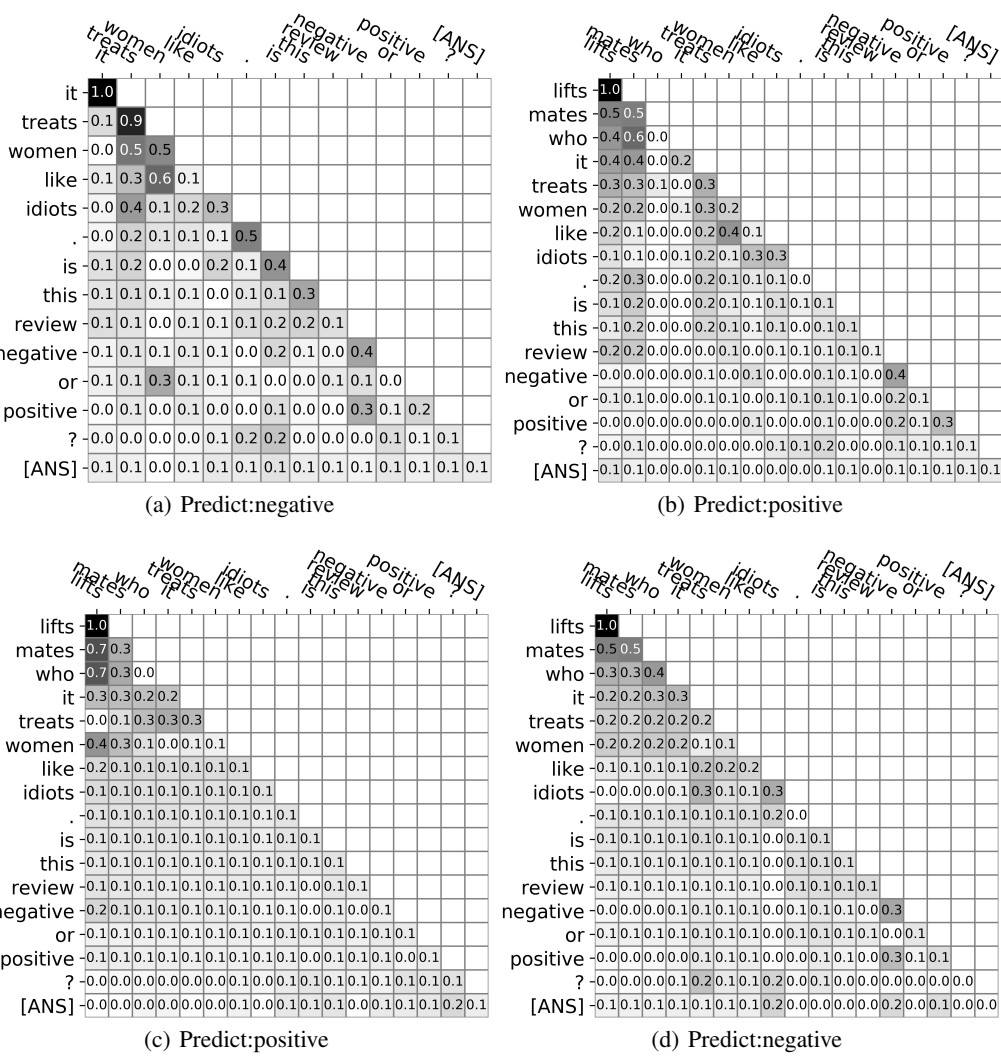

Figure 11: Visualized importance matrices for the second example in Table 17. (a): Original input with original prefix-tuning; (b): UAT-attacked input with original prefix-tuning; (c) and (d): UAT-attacked input with robust prefix-tuning of (c) $N = 24$ and (d) $N = 3$.

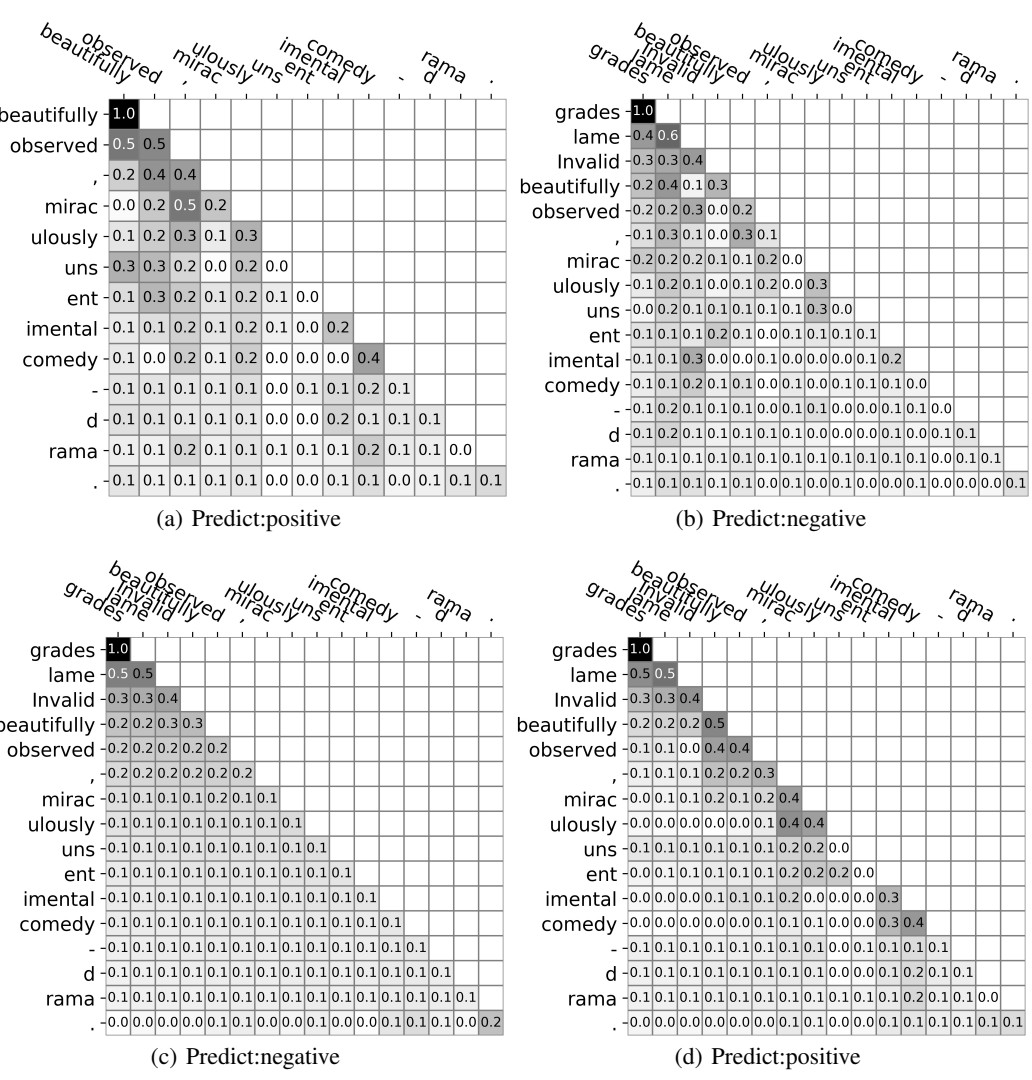

Figure 12: Visualized importance matrices for the third example in Table 17. (a): Original input with original prefix-tuning; (b): UAT-attacked input with original prefix-tuning; (c) and (d): UAT-attacked input with robust prefix-tuning of (c) $N = 24$ and (d) $N = 3$. We omit the visualization of the question part for this example due to its context length.

### F.2.3 QUANTITATIVE ANALYSIS

In this section, we conduct quantitative analysis to show that for UAT, the robust prefix steers the LM to ignore the distraction of the adversarial triggers. We first investigate whether the trigger is assigned with less importance under the robust prefix. Based on the proxy, at each time step $i$, we get the rankings $K_{ij}$ for all visible tokens $x_j$:

$$K[i, : i + 1] = \text{argsort}(I[i, : i + 1]) \tag{19}$$

where the argsort operator, provided in frameworks like NumPy (Harris et al., 2020) and PyTorch (Harris et al., 2020), returns the ranking of each value in the vector by increasing order. Our quantitative analysis is based on token importance rankings at each time step, as the proxy $I$ does not provide a gold indicator for importance value but rather an alternative ordering method. We investigate the importance of the adversarial trigger by calculating the relative ranking of the token with maximal importance in the trigger. We set the trigger length as 3 in our experiments, so for each time step $i \geq 3$, the relative ranking is

$$\max(K[i, : 3])/(i + 1) \times 100\%. \tag{20}$$

The example-level Degree of Distraction (DoD) is defined as the importance of the adversarial trigger averaged at all time steps since the generation of the first token in context:

$$\text{DoD}(x) = \frac{1}{o - 3} \sum_{i=3}^{o} \max(K[i, : 3])/(i + 1) \times 100\% \tag{21}$$

where $x$ represents the input example and $K$ is the calculated importance ranking matrix. Corpus-level Degree of Distraction (cDoD) is then defined as the example-level degree of distraction averaged over all UAT-attacked examples. We calculate the cDoD of the original prefix-tuning as well as our framework with $N = 24$ and $N = 3$ under UAT attack on the SST-2 development set. The results are listed in Table 18. Similar to a trustworthy evaluation for machine translation (Koehn,

Table 18: Calculated corpus-level Degree of Distraction of the original prefix-tuning as well as our framework under UAT attack. Results with † are statistically significant over the that of the standard prefix-tuning baseline with $p < 0.001$.

| Method | cDoD ↓ (%) |
|---|---|
| std. prefix-tuning | 67.67 |
| ours, $N = 24$ | 62.06† |
| ours, $N = 3$ | 61.68† |

2004; Marie et al., 2021), we also conduct statistical significance tests by bootstrapping in the sets of DoD of all test samples calculated for different methods. According to Table 18, the cDoD results of our framework (both $N = 24$ and $N = 3$) are lower than that of the original prefix-tuning with statistical significance ($p < 0.001$).

We proceed to study whether the robust prefix manages to steer the LM to assign the highest importance to the most essential token as the original prefix-tuning does with inputs without adversarial triggers. Let $I_0$ be the importance matrix for the baseline method with clean input, and $I_u$ be the importance matrix for some method with the same input but prepended with adversarial trigger tokens. Let the notation $o$ indicate the output position for the UAT-attacked input, then the output position for the clean input is $o - 3$ (3 is the trigger length). At each time step $i$, We use the indicator

$$E_u(i) = \mathbb{1}\left(\text{argmax}\left(I_0[i, : i + 1]\right) = \text{argmax}\left(I_u[i + 3, : i + 4]\right) - 3\right), \ 0 \leq i \leq o - 3 \tag{22}$$

to determine whether the method with UAT-attacked inputs recognize the essential token by assigning the highest importance to the same token ($\text{argmax}\left(I_u[i + 3, : i + 4]\right) - 3$, left-shifted by the trigger length 3) as the baseline with clean inputs does ($\text{argmax}\left(I_0[i, : i + 1]\right)$). The subtraction of 3 in the RHS of Eq. (22) is due to the right shift of the original input by the trigger tokens. Similar to the calculation of DoD and cDoD, we compute the example-level recognition of the essential token (RoE) for different methods under UAT attack by averaging $E_u(i)$ over all time steps:

$$\text{RoE}(x) = \frac{1}{o - 3} \sum_{i=0}^{o-3} E_u(i). \tag{23}$$

Table 19: Calculated corpus-level Recognition of the Essential token of the original prefix-tuning as well as our framework under UAT attack. Result with † is statistically significant over that of the standard prefix-tuning baseline with $p < 0.001$. Result with ‡ is statistically significant over that of our framework with $N = 24$ with $p < 0.001$.

| Method | cRoE $\uparrow$ (%) |
|---|---|
| std. prefix-tuning | 48.05 |
| ours, $N = 24$ | 34.60 |
| ours, $N = 3$ | 71.63[†‡] |

We then compute the corpus-level Recognition of the Essential token (cRoE) by averaging RoE over all test samples. We calculate the cRoE of the original prefix-tuning as well as our framework with $N = 24$ and $N = 3$ under UAT attack on the SST-2 development set.

The results are listed in Table 19. It can be seen that our framework with $N = 3$ achieves the highest cRoE score, which is also statistically significant compared with the standard prefix-tuning baseline and our framework with $N = 24$. We also find that the cRoE score of our framework with $N = 24$ is lower than that of the baseline. As a result, though the cDoD score shows that our framework with $N = 24$ ignores the distraction of the adversarial triggers to some extent, it seems that the robust prefix with $N = 24$ fails to steer the LM to assign the highest importance to the most essential token under UAT attack. In contrast, when the LM is attacked by UAT, the robust prefix-tuning with $N = 3$ steers the LM so that it not only ignores the distraction from adversarial triggers, but is also (very much likely to be) able to assign the highest importance to the most essential token as the baseline method does with the input without adversarial trigger tokens.

The results from Tables 18 and 19 provide an explanation to the performance shown in Figure 3-(a) that under UAT attack, our framework with $N = 3$ substantially outperforms that with $N = 24$, and both of them achieve higher robustness compared with the standard prefix-tuning baseline. We hope that our quantitative studies can better support the observed behavior and provide a deeper understanding of the robust prefix-tuning framework.

## G  INTERPRETATION FROM THE OPTIMAL CONTROL PERSPECTIVE

In this section, we provide an interpretation from the optimal control perspective for prefix-tuning as well as our robust prefix-tuning framework. Prefix-tuning seeks $P_\theta[i, :]$ with forward propagation of the $j$-th layer of the $L$-layer LM at the output position $o$ as

$$h_o^{(j+1)} = \text{LM}_\Phi^{(j)} \left( h_o^{(j)} \,\middle|\, h_{<o}^{(j)} \right) \tag{24}$$

with $h_i^{(j)} = P_{\theta^{(j)}}[i, :]$ for all $j = 0$ to $L - 1$, $i \in \text{P}_{\text{idx}}$ and $h_i^{(0)} = z_i$ for $i \notin \text{P}_{\text{idx}}$ to optimize

$$\min_{\{\theta^{(0)}, \ldots, \theta^{(L-1)}\}} \mathbb{E}_{(x,y) \sim \mathcal{D}_{tr}} \left[ S \left( h_o^{(L)}, y \right) + \sum_{j=0}^{L-1} R \left( \theta^{(j)} \right) \right], \tag{25}$$

with $S$ as the softmax scoring function, $R$ as the regularizer, $\Phi$ as the LM parameters, $z_i$ as the $i$-th token in the input and $y$ as the label. According to Li et al. (2017), the $S$ and $R$ can be viewed as the terminal and the running loss with $P_\theta$ as the control variables, and the forward and backward propagation of prefix-tuning are equivalent to the calculation of co-state process in Pontryagin's Maximum Principle (Kopp, 1962). Therefore, prefix-tuning can be formalized as seeking the optimal control of the pretrained models for specific downstream tasks.

Similarly, Our robust prefix-tuning framework seeks $P'_\psi[i, :]$ with $h_i^{(j)} = P_{\theta^{(j)}}[i, :] + P_{\psi'^{(j)}}[i, :]$ for all $j = 0$ to $L - 1$, $i \in \text{P}_{\text{idx}}$ in the forward propagation of Eq. (24). Our goal is given by

$$\min_{\{\psi^{(0)}, \ldots, \psi^{(L-1)}\}} \mathbb{E}_{(x,y) \sim \mathcal{D}_{te}} \sum_{j=0}^{L-1} \ell \left( \psi^{(j)}, Q^{(j)} \right), \tag{26}$$

where the running loss $\ell$ is given by Eq. (8) for $j \leq N - 1$ and set to 0 for $j \geq N$. Compared with Eq. (25), there is no terminal loss in Eq. (26) as the label is unknown during test phase. As

suggested by Chen et al. (2021), our robust prefix-tuning can be formalized as seeking the close-loop control for robust downstream tasks.

Before using optimal control theory to prove the above formalizations, we first provide a brief introduction to Li et al. (2017) that reveal the relationship between the optimal control theory and deep learning. We borrow the theorems in Section 4 of Li et al. (2017) and directly follow their notations:

**Theorem G.1.** *(discrete-time PMP) Consider the discrete-time control problem*

$$\min_{\{\theta_0,...,\theta_{T-1}\}\in\Theta^T} \Phi(x_T) + \delta \sum_{t=0}^{T-1} L(\theta_t), \tag{27}$$

$$x_{t+1} = x_t + \delta f_t(x_t, \theta_t), \ x_0 = x, \ 0 \le t \le T-1$$

*where $\Phi$ is the termination loss and $L$ is the running loss. Then there exists a co-process*

$$x_{t+1}^* = g_t(x_t^*, \theta_t^*), \qquad x_0^* = x, \tag{28}$$

$$p_t^* = \nabla_x H_t(x_t^*, p_{t+1}^*, \theta_t), \quad p_{T+1}^* = -\nabla_x \Phi(x_{T+1}^*) \tag{29}$$

*such that*

$$H_t(x_t^*, p_{t+1}^*, \theta_t^*) \ge H_t(x_t^*, p_{t+1}^*, \theta), \ \theta \in \Theta, \ 0 \le t \le T-1. \tag{30}$$

*Here $g_t(x_t, \theta_t) := x_t + \delta f_t(x_t, \theta_t)$ and*

$$H_t(x, p, \theta) = p \cdot g_t(x, \theta) - \delta L(\theta) \tag{31}$$

*is the scaled discrete Hamiltonian.*

**Theorem G.2.** *(discrete-time MSA) The co-process in Theorem G.1 can be determined by applying the discrete-time method of successive approximations (MSA). For each iteration $k$,*

*set $x_0^k = x$, and*

$$x_{t+1}^k = g_t(x_t^k, \theta_t^k) \tag{32}$$

*with $t$ enumerating from 0 to $T-1$;*

*then set $p_T^k = -\nabla_x \Phi(x_T^k)$, and*

$$p_t^k = \nabla_x H_t(x_t^k, p_{t+1}^k, \theta_t^k) \tag{33}$$

*with $t$ enumerating from $T-1$ to 0;*

*finally, with $t$ enumerating from 0 to $T-1$, set*

$$\theta_t^{k+1} = \theta_t^k + \eta \nabla_\theta H_t(x_t^k, p_{t+1}^k, \theta_t^k). \tag{34}$$

**Theorem G.3.** *(equivalence between MSA and back-propagation) The MSA in Theorem G.2 is equivalent to gradient descent with back-propagation.*

The proofs of Theorems G.1, G.2 and G.3 are provided in Li et al. (2017).

We now investigate the forward propagation of prefix-tuning. In Eq. (24), $\text{LM}_\Phi^{(j)}$, the $j$-th layer of the LM, can be decomposed into a self-attention layer ($\text{SAN}_\Phi^{(j)}$) and a FFN layer ($\text{FFN}_\Phi^{(j)}$). Formally,

$$h_o^{(j+1)} = h_o^{(j)} + \text{SAN}_\Phi^{(j)}\left(h_o^{(j)}, h_{<o}^{(j)}\right) + \text{FFN}_\Phi^{(j)}\left(h_o^{(j)} + \text{SAN}_\Phi^{(j)}\left(h_o^{(j)}, h_{<o}^{(j)}\right)\right) \tag{35}$$

with $h_i^{(j)} = P_{\theta^{(j)}}[i,:]$ for $i \in \text{P}_{\text{idx}}$. As Eq. (35) is recursive and according to the fact that the pretrained LM is autoregressive, after unrolling the recursion for all $h_{<o}^{(j)}$ in Eq. (35), $h_o^{(j+1)}$ can be represented in the form of

$$h_o^{(j+1)} = h_o^{(j)} + \mathcal{G}_\Phi^{(j)}\left(h_o^{(j)}, \theta^{(j)}\right). \tag{36}$$

Now we take Eq. (25), the objective of prefix-tuning, into consideration. The optimization problem is now formulated as

$$\min_{\{\theta_0,...,\theta_{L-1}\}\in\Theta^L} \mathbb{E}_{(x,y)\sim\mathcal{D}_{tr}} \left[ S\left(h_o^{(L)}, y\right) + \sum_{j=0}^{L-1} R\left(\theta^{(j)}\right) \right] \tag{37}$$

$$h_o^{(j+1)} = h_o^{(j)} + \mathcal{G}_\Phi^{(j)}\left(h_o^{(j)}, \theta^{(j)}\right), \ h_o^{(0)} = z_o = [\text{ANS}], \ 0 \le j \le L-1.$$

Using Theorem G.1, we know that the objective of prefix-tuning can be formulated as a discrete-time control problem. We then use the MSA described in Theorem G.2 to determine the co-process that solves the control problem and realize that the MSA is equivalent to back-propagation with Theorem G.3, which recovers the training phase of prefix-tuning. As a result, we can conclude that prefix-tuning seeks the optimal control of the pretrained models for specific downstream tasks.

Our robust prefix-tuning framework can also be formalized as the close-loop control of the obtained prefix for robust downstream tasks. According to Chen et al. (2021), the close-loop control framework for robust neural networks is defined as

$$
\min_{\{\pi_0,\ldots,\pi_{L-1}\}} \sum_{t=0}^{L-1} \ell(x_t, \pi_t(x_t)),
$$
$$
x_{t+1} = f_t(x_t, \pi_t(x_t)),\ x_0 = x,\ 0 \le t \le L-1. \tag{38}
$$

where $\ell(x_t, \pi_t(x_t))$ is the running loss. According to Eq. (26), the goal of our robust prefix-tuning framework can be formulated into the form

$$
\min_{\{\psi^{(0)},\ldots,\psi^{(L-1)}\}} \mathbb{E}_{(x,y)\sim\mathcal{D}_{te}} \sum_{j=0}^{L-1} \mathcal{R}^{(j)}_{\Phi,\theta^{(j)}}\left(h_o^{(j)}, \psi^{(j)}\left(h_o^{(j)}\right)\right),
$$
$$
h_o^{(j+1)} = h_o^{(j)} + \widetilde{\mathcal{G}}^{(j)}_{\Phi,\theta^{(j)}}\left(h_o^{(j)}, \psi^{(j)}\left(h_o^{(j)}\right)\right),\ h_o^{(0)} = z_o = [\text{ANS}],\ 0 \le j \le L-1. \tag{39}
$$

Since Theorems G.1, G.2 and G.3 can also be applied to Eq. (39), the optimization of our robust prefix is equivalent to seeking the close-loop control for robust downstream tasks.

In conclusion, recent years have witnessed the paradigm shift in NLP from *adapting LMs to downstream tasks* to *adapting downstream tasks to LMs* (Liu et al., 2021). In this work, we provide the interpretation of prefix-tuning as well as our robust prefix-tuning framework from the optimal control perspective. We hope that our work can bring insight to the future design of robust and efficient finetuning approaches for large-scale pretrained models.

