# OpenReview forum: "On Robust Prefix-Tuning for Text Classification"
_ICLR.cc/2022/Conference — ICLR 2022 Poster_

### Official Review · Reviewer_GPh4 · 2021-10-17

**Correctness:** 4
**Technical Novelty And Significance:** 4
**Empirical Novelty And Significance:** 4
**Recommendation:** 6
**Confidence:** 2

**Main Review:**

Strengths
- The problem being addressed is cutting edge (we are barely understanding prompt tuning, and this paper already jumps ahead to adversarial defenses!)
- The approach seems novel, though I haven't searched for related literature carefully.
- Clear logic motivating the problem and what constraints need to be accounted for while solving it.
- Clear research question.

Weaknesses
- The experiments are OK but could be a bit more streamlined. I guess two things came up when I was looking at them.
	- It may be good to expand the scope of the paper to generation tasks, as these are likely more suceptible to adversarial attacks. What I mean by this is that the worse case scenario is much worse for generation tasks: while in binary classification the worse case is bad accuracy, for adversarial attacks on generation tasks, potentially very harmful text could be generated, which is much worse than simply getting the answer wrong.
	- The other thing is that it is hard to contextualize how good the numbers in your framework are. For example, I have no idea how good 52% on VIPR for SST2 is. If previously proposed defenses are able to get 90% on the same task, then it is unlikely that the proposed method will gain traction, even if it is more computationally efficient with prefix tuning. So it would strengthen this paper a lot to have this comparison.

Other comments
- I think the paper may benefit from spending just a little more time describing how susceptible prefix tuning is to adversarial attacks compared with regular finetuning. It seems like it is very susceptible, especially for easy tasks like text classification.
- Appreciate the candidness in Figure 2 in showing that adversarial training takes longer.
- It will be good to see if the proposed method not only improves performance against adversarial attacks, but also if it improves performance against different paraphrases of expressing the context and question, for example.

Minor points
- The grammar of the paper will benefit from having a native english speaker proof-read it. E.g., "as well as remaining the pretrained models unmodified" can be re-written as "without modifying the pretrained model parameters".
- Since a lot of work was put into the paper, I will pay my respect and make a picky point: the bibliography could be cleaned up a bit. E.g., capitalize RoBERTa correctly, add URLs consistently, capitalize all conference names (e.g., IEEE transactions).

**Summary Of The Paper:**


The paper is a focused contribution at the intersection of defending against text attacks and prompt tuning.
The paper requires the reader to understand the context and motivation of several different things before understanding the contribution of the paper. First, adversarial examples can attack a text classifier, such as UAT. Second, various techniques defend against these attacks in different way. However, these techniques requiring modifying the parameters of the LM or other additional computational burdens. These techniques can be used with prompt tuning, but then the benefits of prompt tuning go away. Hence, there ought to be a technique that improves the robustness of prompt tuning without removing its benefits over regular finetuning.
The paper proposes such a technique and do experiments for three text classification tasks and various adversarial attacks.

**Summary Of The Review:**


The paper studies the very timely topic of adversarial attacks against prefix tuning. The paper proposes a method that maintains the advantages of prefix-tuning against finetuning, a formidible problem. The experiments are OK as of now, though expanding the scope to generation tasks could make the impact substantially larger. The characterization of the method could also be further improved by providing finetuning (with and without defenses) as baselines. Overall, I lean towards acceptance, though I am not an expert in adversarial attacks/defenses.

---

> ### Author Response · Authors · 2021-11-23
> **Response [4.1] to Reviewer GPh4**
>
> We'd like to thank you for the positive feedback and acknowledgment of novelty! We address the comments and suggestions below.
>
> [4.1] Expand the scope to generation tasks, as generating harmful text is much worse than getting the classification answer wrong.
>
> Thanks for the suggestion. We totally agree that generalizing our method to generation tasks (such as detoxification) will be both important and impactful. However, non-trivial adaptations need to be made in order to generalize our framework to text generation, as there are several differences between text classification and generation tasks.
>
> The most important adaptation to generation tasks is to determine the output position $o$ where the activations are collected (the first step in our framework; see Section 3). For text generation, the $y = [label]$ is a ground-truth sentence, with often $|y| > 1$; during inference, several consecutive tokens will be generated autoregressively by the LM.
> This is different from text classification tasks where $|y|=1$ and only a single token will be generated as the label prediction during inference (see Section 2). Consequently, it requires further study that at which position(s) the activations should be collected so that the LM can be rectified/detoxified. A primary solution is to collect the activations at the position of the first generated token only during inference, but we believe there are better solutions that will achieve stronger performance but also require more research.
>
> Given the time limit, we focus on adversarial defense against text classification in this work. We've included your suggestion in Section 6 as an important future work ("extending our framework to text generation").

---

> > ### Author Response · Authors · 2021-11-23
> > **Responses [4.2]-[4.3] to Reviewer GPh4**
> >
> > [4.2] Contextualize the results with previously proposed defenses.
> >
> > Thanks for the suggestion. The type of previously proposed defenses that enables a fair comparison with our setting is adversarial training, which has been studied and compared in Sections 4.2 and 4.3, as well as Appendices C and D. While other types of defenses are not our focus as they will remove the benefits of prefix-tuning, we believe the concern lies in whether there are sufficiently strong defenses that have already achieved substantial robustness results on the tasks.
> >
> > We consider defenses for pretrained language models and their performance for contextualization.
> > - For the UAT attack, to the best of our knowledge, no defense techniques have been proposed on large-scale pretrained LM (see the discussion in the third item in Appendix 1).
> > -For in-sentence attacks,
> >  - [1] use BERT as the pretrained LM with multiple different defenses on the AG News task. The benchmarked SOTA method (FreeLB++ for BERT) achieves 95.1% on clean test data and 55.9% on TextBugger. In contrast, our framework with adversarial prefix-tuning achieves 90.26% on clean test data and 54.66% on TextBugger.
> >  - [2] use BERT and RoBERTa as the pretrained LMs with multiple different defenses on the SST-2 task. The proposed method for BERT achieves 91.54% on clean test data and 38.82% on PWWS. The proposed method for RoBERTa achieves 93.96% on clean test data and 41.85% on PWWS. In contrast, our framework with adversarial prefix-tuning achieves 93.79% on clean test data and 57.55% on PWWS.
> >
> > Based on the facts, we aim to provide a basic understanding of the tasks on how much progress has been made in terms of adversarial robustness. First of all, our framework is the first to defend against UAT on large-scale pretrained LM. Second, under in-sentence attacks, our robustness results are also promising in general compared to the listed results. Finally, prefix-tuning might underperform regular finetuning in terms of clean accuracy when the scale of training data is large, as the optimization of the small amount of prefix parameters becomes difficult. This explains that in the result numbers listed above, our performance on clean test data is surpassed in AG News but is comparable in SST-2, because the size of the training data in AG News is 115,000, while that in SST-2 is 6,920 (see Table 10). It is also proposed that using a larger-scale LM is helpful to "close the gap" between continuous prompt tuning and regular fine-tuning[3], but it is beyond the scope of this work.
> >
> > We also have to add a reminder that the numbers listed above do not form fair comparisons as both the selection of the pretrained LM (BERT/RoBERTa v.s. GPT-2 medium) and the tuning techniques (regular finetuning, updating the entire LM v.s. prefix-tuning, updating only the prefix embedding parameters) are different. We've also conducted experiments that enable a relatively more fair comparison: see [4.3].
> >
> > [4.3] Describe how susceptible prefix tuning is to adversarial attacks compared with regular finetuning; provide finetuning (with and without defenses) as baselines.
> >
> > Thanks for the suggestion. We've provided regular finetuning baselines for comparison using GPT-2 medium as the pretrained LM, and preprocessing the data following the protocol of decaNLP, which is the same as that used in prefix-tuning. The comparison of the accuracy and the robustness performance between (both standard and adversarial) prefix-tuning and full tuning is shown in Table 6. The detailed comparison between prefix-tuning and full tuning is attached in Appendix A; also see the response to Comment [2.1] by Reviewer iz2s for the reference of challenges on robustness for prefix-tuning and contributions of our robust prefix-tuning framework.
> >
> > References:
> >
> > [1] Chenglei Si, Zhengyan Zhang, Fanchao Qi, Zhiyuan Liu, Yasheng Wang, Qun Liu, and Maosong Sun. Better Robustness by More Coverage: Adversarial and Mixup Data Augmentation for Robust Finetuning. Findings of ACL 2021.
> >
> > [2] Zongyi Li, Jianhan Xu, Jiehang Zeng, Linyang Li, Xiaoqing Zheng, Qi Zhang, Kai-Wei Chang, and Cho-Jui Hsieh. Searching for an Effective Defender: Benchmarking Defense against Adversarial Word Substitution. EMNLP 2021.
> >
> > [3] Brian Lester, Rami Al-Rfou, Noah Constant. The Power of Scale for Parameter-Efficient Prompt Tuning. EMNLP 2021.

---

> > > ### Author Response · Authors · 2021-11-23
> > > **Responses [4.4]-[4.6] to Reviewer GPh4**
> > >
> > > [4.4] Observe whether the proposed method improves performance against different paraphrases of expressing the context and question.
> > >
> > > Thanks for the suggestion. In fact, we've already evaluated our method against the SCPN attack that paraphrases the context part. According to Section 2, the context part is the real input sentence to be classified, while the question part is just a fixed sequence concatenated during data preprocessing following the protocol of decaNLP. According to the results, our method consistently improves performance against the paraphrasing of the context as well.
> > >
> > > [4.5] Proofread the grammar of the paper.
> > >
> > > Thanks for the suggestion. We've improved the writing in this revision.
> > >
> > > [4.6] Clean up the bibliography.
> > >
> > > Thanks for the suggestion. We've cleaned up the bibliography in this revision.

---

### Official Review · Reviewer_rkLx · 2021-10-28

**Correctness:** 2
**Technical Novelty And Significance:** 3
**Empirical Novelty And Significance:** 3
**Recommendation:** 6
**Confidence:** 4

**Main Review:**

Review note: I am not familiar enough with Optimal Control Theory to evaluate the soundness of Section 6 / Appendix E. I will leave it outside the scope of my review and hope other reviewers can fill in the gap.

Pros:
- The results are quite convincing in nearly all settings (aside from important caveats in cons). While some scores are still quite low after RPT, they are consistently better than the baselines. Notably, the method can provide additional gains when used with other defense methods, such as adversarial data augmentation and adversarial training.
- The need for this method is well-motivated
- The paper is good at emphasizing different priorities rather than only the end accuracy. For instance, time to accuracy in Fig. 2
- Although there are some questions about the validity of using attention weights (see https://arxiv.org/abs/1902.10186 which likely should be mentioned as a caveat), I found Section 5 insightful.

Cons:
- Inference Batch, batch size and its importance:
The batch size at inference seems like an important variable. Indeed, my understanding of Section 3 and Eq. 8 is that the specific batches used at inference will play an important role as they impact P_psi. This is my **key concern with this paper** as it brings up quite a few issues that should be mentioned.
  - Inference for a datapoint depends also on other datapoints in the batch. This is different enough from other ML setups that it should be highlighted. It also causes reproducibility issues. For instance, batch norm avoids this by fixing batch statistics at inference.
  - It is not clear that the method works for low inference batch size (opt of Eq. 8). The inference batch size used is not mentioned anywhere. If the method requires batch size > some N (or if performance varies widely with batch size), this seems a strong assumption that should be made clear. You do not always get several samples of the attack on your system.
  - It is not clear how well the method works when the inference batches are a mix of unperturbed samples and perturbed ones. This seems like a more realistic attack scenario.
  - It is not clear how well the method works if there are different attacks in the inference batch, which also seems like a more realistic threat model.

Overall, I feel like some answers to the above would diminish my concerns, most notably:
  - What is the inference batch size used in the experiments?
  - How does performance vary with inference batch size (one or two settings should be enough)
  - Does the method work when only x of N samples in the test batch are adversarial? Does the example work when they are two types of attack

Dataset statistics and mode prediction

It would be helpful to remind the reader for each dataset how many classes they are / what the mode/random pred accuracy is to help interpret results. For instance, on Table 1 under PWWS improves from 16.64 to 50 for SST-2 and 25 to 34 on SNLI but that is the same as the accuracy of a random predictor.

Writing:

The writing could be improved substantially. Some examples: p2 “remaining the pretrained models unmodified” -> “keeping”. Also on p3 “remaining its lightweightness”. I suggest doing another pass as it does not align with the quality of the rest of the paper.

Related work:

I feel like other methods for Parameter efficient transfer learning, such as Adapters, normal prompting should be quickly mentioned. Same goes for comparable approaches such as P-tuning.
As mentioned earlier, I would caveat Section 5 with discussions of the validity of using attention for explanation, such as the Attention is Not Explanation paper.

Edit:

Given the author's response, I am raising my score slightly. There are still some concerns over the threat model but some of my questions on the impact of test batch size have been answered.



**Summary Of The Paper:**


This paper introduces a tweak to Prefix-Tuning to make it more resilient to adversarial perturbations of the input. The idea is to add a batch-level prefix at inference to the original one which enhances robustness. Critically, Robust Prefix-Tuning (RPT) does not require auxiliary model updates or storage, in contrast with other robustness methods.Thus, this approach makes prefix-tuning more robust while preserving its modularity and low storage requirements.

The authors conduct experiments on 3 text classification tasks, 5 textual attacks and different training regimes (normal training, adversarial training and adversarial data augmentation). In nearly all instances, their method improves robustness (sometimes considerably so) while preserving the accuracy on the original text.

The authors also present RPT from an optimal control perspective and conduct a qualitative study that shows how RPT impacts attention weights.


**Summary Of The Review:**

The motivation, experimental settings and results look good overall.

One major caveat however is that it is not clear how flexible the inference setup is. This is critical as the predictions for a datapoint at inference depend on the other datapoints in the batch. Currently, it seems like the inference is being done with a batch of all the same attack. These assumptions are simply not realistic as a threat model. Without understanding how this performs under a more realistic threat model, I cannot score this paper higher. I have highlighted experiments that would provide more realistic results.
Despite my "marginally below" score, I do not think the paper can be accepted without an answer on this point. Reject is too harsh for a paper that is otherwise promising.

The paper would also benefit from another writing pass, mentioning missing related work and clarifying some properties of the dataset.

---

> ### Author Response · Authors · 2021-11-23
> **Responses [3.1]-[3.7] to Reviewer rkLx**
>
> We'd like to thank you for your positive feedback and acknowledgment of our work! We are also grateful that you remind us of the importance of the inference batch size. We now address the comments and suggestions below.
>
> [3.1] What is the inference batch size used in the experiments?
>
> Thanks for the question. We set a fixed ratio of the GPU memory for loading data and use an adaptive test batch size (in terms of the number of sequences in a test batch) to make full use of GPU memory in the experiments. It turns out that the total number of tokens in a test batch is roughly a constant. For example, the batch size (number of tokens) in SST-2 experiments is roughly 1,700. During inference, The sizes of the test batches are 106 sequences of 16 tokens, 94 sequences of 18 tokens, 89 sequences of 19 tokens, etc. We've provided more details on this for reproducibility in Appendix B. The techniques might have improved the inference efficiency, but we are really grateful that you remind us that the number of sequences in a test batch affects the results. We've conducted additional experiments with the number of test sequences in a test batch fixed. See [3.2]:
>
> [3.2] How does performance vary with different (low) inference batch sizes?
>
> Thanks for the question. We conducted experiments where the test batch size (number of sequences) is fixed as 1, 2, 4 on SST-2 for the ablation study shown in Section 4.5. It is shown that our framework still substantially improves the robustness of prefix-tuning with different small test batch sizes. We have also discussed the effect of small test batch sizes in depth in Appendix E.1. We have also provided experimental results on SST-2 and AG based on both standard and adversarial prefix-tuning with test batch size (number of sequences) = 1 in Table 13. In all experiments, our framework achieves substantially stronger results than the baselines.
>
> [3.3] Does the method work when only x of N samples in the test batch are adversarial? Does the example work when they are two types of attack?
>
> Thanks for the question. The questions are challenging, as previously proposed defenses demonstrate robustness improvement by reporting only accuracy under specific attack [1][2][3]. However, we also agree that these are the most realistic settings. We conducted additional experiments that evaluate the performance of RPT when the test data is mixed with original and perturbed inputs and when they are two types of attack. Results show that our framework also works well in these settings. The details are attached in Section 4.5 and Appendix E.2.
>
> [3.4] Clarify some properties of dataset statistics such as the number of classes.
>
> Thanks. We've added more properties of dataset statistics in Table 10.
>
> [3.5] Improve the writing.
>
> Thanks. We've improved the writing in this revision.
>
> [3.6] Mention related work.
>
> Thanks. In fact, we mentioned some related work about soft prompt-based tuning methods at the end of Section 3 and discussed how to generalize our framework to them. We have also cited Adapter and other normal prompt tuning approaches in Section 1. We have also added more references in this revision.
>
> [3.7] Add caveats to Section 5 with discussions of the validity of using attention for explanations.
>
> Thanks. We have added caveats in Section 5.1 and Appendix F.1 for the analysis of in-sentence attacks that the analysis based on attention is of the final layer in the LM only. For UAT attack, we use an alternative proxy for visualization and quantitative studies, so that the analysis is more reliable (Section 5.2, Appendices F.2.1, F.2.2, F.2.3). See responses to Comments [1.3] and [1.4] by Reviewer yKwP for details.
>
> References:
>
> [1] Xinshuai Dong, Anh Tuan Luu, Rongrong Ji, and Hong Liu. Towards Robustness Against Natural
> Language Word Substitutions. ICLR 2021.
>
> [2] Chenglei Si, Zhengyan Zhang, Fanchao Qi, Zhiyuan Liu, Yasheng Wang, Qun Liu, and Maosong
> Sun. Better Robustness by More Coverage: Adversarial and Mixup Data Augmentation for Robust Finetuning. Findings of ACL 2021.
>
> [3] Zongyi Li, Jianhan Xu, Jiehang Zeng, Linyang Li, Xiaoqing Zheng, Qi Zhang, Kai-Wei Chang,
> and Cho-Jui Hsieh. Searching for an Effective Defender: Benchmarking Defense against Adversarial Word Substitution. EMNLP 2021.

---

> > ### Comment · Reviewer_rkLx · 2021-11-26
> > **Thanks for the update, revised score slightly upwards**
> >
> > Hi,
> >
> > Thanks for your answers, esp. the new experiments in Section 4.5 and Appendix E.
> >
> > I am somewhat reassured by the answer to "How does performance vary with different (low) inference batch sizes?". Though the variance in results can be quite high and it will be surprising for users to have test performance depend on batch size, the results are not catastrophically worse when all the samples are from the same attack.
> >
> > Regarding the "mixed" attack scenarios, thanks for investigating this. However, I think the most interesting case is when test batch size is not 1 as you are otherwise simply sidestepping the issue. This is the most realistic attack scenario / threat model.
> >
> > Overall, I feel like the authors have done a decent attempt at alleviating some (though not all) of my concerns on the impact of batch size. I am raising my score slightly. I hope the authors will investigate the batch size > 1, mixed data scenario in their camera-ready release.

---

> > > ### Author Response · Authors · 2021-11-27
> > > **Added mixed data experiments with test batch size > 1, results still strong**
> > >
> > > Hi,
> > >
> > > Thank you for your reply and acknowledgment! We want to update our new experiments with the setting of mixed test data and test batch size > 1.
> > >
> > > First, we would like to explain the reason why we used test batch size = 1 in the experiments for Table 4 and Appendix E.2. When a threat model attacks a system, it perturbs the test input so that the system is fooled. The threat model, however, cannot modify the settings (hyperparameters, including the test batch size) used in our (defense) system. Therefore, we set test batch size = 1 so that the scenario where there are mixed data in a batch can be sidestepped. As a result, when a mixed test data batch is sent from the threat model, the batch will be broken into a sequence of single test datum, which will be processed by our defense model sequentially (as we set test batch size = 1).
> > >
> > > But we also realize that it is the most interesting case. What's more, our framework will be more efficient and more practical if it is able to defend against mixed test data with test batch size > 1. We conducted mixed data experiments with the adaptive test batch size (introduced in [3.1] and Appendix B). We augment Table 15 with the new results, listed as below:
> > >
> > > | Benchmark | Method                       | C + B | C + P | V + B | V + P | S + B | S + P | U + B | U + P |
> > > |-----------|------------------------------|:-----:|:-----:|:-----:|:-----:|:-----:|:-----:|:-----:|:-----:|
> > > | SST-2     | adversarial prefix-tuning    | 60.65 | 62.27 | 17.76 | 20.43 | 29.57 | 31.85 | 18.12 | 20.81 |
> > > |           | ours, mixed, bsz=1           | **69.71** | **71.11** | 46.02 | 47.23 | 55.63 | 56.92 | 70.04 | **71.67** |
> > > |           | ours, mixed, bsz adaptive    | 68.12 | 68.75 | **47.47** | **47.31** | **58.51** | **59.45** | **71.14** | 71.25 |
> > > |           | ours, separate, bsz=1        | 74.66 | 75.40 | 50.66 | 51.40 | 57.75 | 58.49 | 71.28 | 72.02 |
> > > |           | ours, separate, bsz adaptive | 75.48 | 75.67 | 50.39 | 50.58 | 58.93 | 59.12 | 74.52 | 74.71
> > > | AG's News | adversarial prefix-tuning    | 52.38 | 53.41 | 26.54 | 27.14 | 34.86 | 35.36 | 40.97 | 42.16 |
> > > |           | ours, mixed, bsz=1           | **71.36** | 71.39 | 42.84 | **43.99** | 49.04 | 50.12 | 69.88 | 71.30 |
> > > |           | ours, mixed, bsz adaptive    | 70.51 | **71.91** | **43.18** | 43.53 | **51.45** | **52.76** | **70.34** | **71.33** |
> > > |           | ours, separate, bsz=1        | 71.77 | 72.76 | 44.04 | 45.03 | 51.59 | 52.58 | 70.46 | 71.44 |
> > > |           | ours, separate, bsz adaptive | 72.46 | 73.36 | 42.96 | 43.85 | 51.35 | 52.24 | 70.96 | 71.86 |
> > >
> > > From the results, we find that with the test batch size > 1, the performance of our framework under most mixing settings is better than that with test batch size = 1. This shows that the normalization before projection can be helpful for the test data under different settings (clean, or perturbed by different attacks).
> > >
> > > We've also attached the results of averaged separate testing with adaptive batch size for comparison. Comparing the results, we find that under some settings (for example, S+P on SST-2), the mixed test with batch size > 1 even achieves slightly better results than the separate test. This suggests that sometimes our framework can even benefit from mixed test data, thanks to the normalization before projection operation as it normalizes the activation among the different types of test data in the mixed batch. Of course, there are also cases when the performance of our framework is affected due to the normalization, especially the results on clean test data (see the "C+B" and "C+P" columns). But for all cases, the results of our framework are always substantially stronger than those of the baselines.
> > >
> > > We will add the detailed analysis in our final version. Thank you so much for the suggestions, as well as the time and effort you spend reviewing our work! We've really benefited a lot.
> > >
> > > Best regards,
> > >
> > > Authors

---

### Official Review · Reviewer_iz2s · 2021-11-02

**Correctness:** 3
**Technical Novelty And Significance:** 3
**Empirical Novelty And Significance:** 3
**Recommendation:** 6
**Confidence:** 3

**Main Review:**

The authors study a novel problem in lightweight fine-tuning methods. Most studies aim to match the performance of full model tuning via updating a subset of parameters but rarely study the robustness of lightweight fine-tuning methods.


Strengths:
1. The authors study an important and novel problem.
2. The authors provide a simple yet effective method with motivations. The proposed method is shown to be effective even combining with adversarial methods.

Weakness:
1. The authors argue that robustness is important for lightweight tuning methods. But I still think it is better to provide a comparison between lightweight tuning methods will full tuning methods. Provide a basic starting observation about whether lightweight methods bring more challenges on robustness or not/



**Summary Of The Paper:**

The paper investigates the robustness of prefix-tuning methods and proposes a simple yet effective method to improve the robustness. The experiments show that the proposed method can largely improve the performance in adversarial settings and slightly improve the performance in clean settings.

**Summary Of The Review:**

The studied problem is important and novel. The proposed method is simple and clear. The experiments justify the effectiveness of the proposed method.

---

> ### Author Response · Authors · 2021-11-23
> **Response [2.1] to Reviewer iz2s**
>
> We'd like to thank you for your positive comments and acknowledgment of our work! We address the comments and suggestions below.
>
> [2.1] Provide a comparison between lightweight tuning methods will full tuning methods. Observe whether lightweight methods bring more challenges on robustness or not.
>
> Thanks for the suggestion. We conducted full tuning experiments for comparison using GPT-2 medium as the pretrained LM, and preprocessing the data following the protocol of decaNLP, which is the same as that used in prefix-tuning. The comparison of the accuracy and the robustness performance between (both standard and adversarial) prefix-tuning and full tuning is provided in Appendix A.
>
> We summarize the challenges on robustness for prefix-tuning and the especial contribution of our robust prefix-tuning framework as follows (also detailed in Appendix A):
>
> - Difficulty in optimization for prefix-tuning. As the number of parameters in a prefix is small, the difficulty in optimizing prefix might have brought challenges on robustness to prefix-tuning. Table 6 shows that the performance of prefix-tuning baselines is surpassed by that of full-tuning baselines under attack.
>
> - Trade-off between space and time. Prefix-tuning is lightweight but might take more epochs to converge (due to the optimization difficulty; also see the dev acc curves in Figure 2-(a) for reference). As a result, an ideal solution should be neither harmful to the benefits of prefix-tuning (lightweightness, modularity, not modifying the LM parameters) nor too slow for fear that the time complexity of prefix-tuning further deteriorates. Our framework serves as a solution that keeps the strengths without weakening the weakness.
>
> - Lack of robustness against UAT. Existing defense approaches against UAT maintain additional adversary detectors and are not tested on pretrained LM due to its large scale. To the best of our knowledge, our framework is the first solution to defend against UAT by tuning an additional prefix based on prefix-tuning for pretrained LM. Positioned before the adversarial triggers, the robust prefix regulates the activation at the output position so that "the distraction can be ignored".

---

### Official Review · Reviewer_yKwP · 2021-11-02

**Correctness:** 3
**Technical Novelty And Significance:** 3
**Empirical Novelty And Significance:** 4
**Recommendation:** 6
**Confidence:** 3

**Main Review:**

Below are the detailed strengths and weaknesses:

*Strengths:*

1. Adversarial robustness is an important problem and has not been explored much for relatively new prefix/prompt tuning approaches. Thus the topic of this paper can be of interest to a general audience.  Also, this paper is timely given the recent attention on prompts.
2. The idea of optimizing the geometry similarity to defend against attacks is interesting and novel from my perspective. Particularly I like test-time tuning which could adapt to different types of attacks on the fly.
3. The experimental results are strong.



~~~~
Updates after Rebuttal: Most of the following concerns have been addressed in the revision, and I have increased my score
~~~~

*Weaknesses*:

1. What is the batch size at test time tuning? Is the added robust prefix the same for the entire test set, or the same within a batch but different across batches, or unique for every test example? This is an important point to assess whether the experiments are in an online setting where test data arrives in-stream or not.
2. Section 5 is not very convincing to me:
(1) there are only several case studies without any quantitative results; I think it may appear in the appendix only or just uses a short paragraph in the main body (because the statements in Section 5 can only be good hypotheses which are not well-supported by quantitative evidence, also, attention is not a convincing proxy for the explanation, see point (2)), yet it takes more than one page in the current version;
(2) the interpretation from the perspective of attention weights bothers me a bit, attention as an explanatory tool is known to not be faithful [1] – a larger attention weight does not necessarily mean the final prediction depends on it more than others, and vice versa, thus maybe not overread it too much. As said above, it is ok to do it this way in the appendix, but using over one page of the main content for this is not convincing to me.
3. In the attention visualization figures (e.g. Figure 4), why does a word attend itself as well instead of only attending to the contexts? For LM, I think that the diagonals in the attention figure should be zero, or am I missing something? Also, the presentation could be improved here, in the figure caption you can explain what the rows and columns mean in the visualization to make it easier to read.
4. Section 6 is difficult to follow without reading the appendix, and it is disconnected from the rest of the paper in terms of the paper structure – having a theoretical interpretation section after experiments at the end of the paper is not a good presentation structure in my view. If you think Section 6 is important to have, move it before the experiment section and clarify more details to make it more self-contained; if it is not very important, you can just put it into the appendix while briefly mentioning it in the main body.
5. Besides the presentation issues above, there are some other minor presentation places which could be improved, e.g.
(1) Eq 4, an undefined variable X_C suddenly comes in without explanation
(2) Better to explain Eq (8) with more text given that this is an important equation for the proposed method

[1] Serrano et al. Is Attention Interpretable? ACL 2019




**Summary Of The Paper:**

This paper focuses on improving the adversarial robustness of prefix tuning (Li et al. 2021), which is a recent parameter-efficient tuning method. Specifically, the paper proposes to add extra batch-level prefixes that are tuned for each test batch on the fly, to minimize the distance between hidden activations of the test samples and the canonical manifold obtained from the hidden activations from correctly classified training samples. The intuition is to optimize the added batch-level prefixes so that the geometry of hidden states from adversarial examples is closer to that of training examples. Experiments on three text classification benchmarks across several different adversarial attacks demonstrate the effectiveness of the method.

**Summary Of The Review:**

There are novel contributions of the paper both technically and empirically, however some major analysis sections are not convincing and a significant portion of the presentation needs to be improved.

---

> ### Author Response · Authors · 2021-11-23
> **Responses [1.1]-[1.8] to Reviewer yKwP**
>
> We'd like to thank you for your positive comments and acknowledgment of our work! We also appreciate the suggestions on improving the interpretation sections. We now address the comments and suggestions below.
>
> [1.1] Is the added robust prefix the same for the entire test set, or the same within a batch but different across batches, or unique for every test example?
>
> Thanks for the question. The added prefix $P'_\psi$ is the same within a batch but different across batches.
>
> [1.2] What is the batch size at test time tuning?
>
> Thanks for the question. Please refer to the responses to Comments [3.1]-[3.3] by Reviewer rkLx for details.
>
> [1.3] In Section 5, the interpretation from the perspective of attention weights is not convincing.
>
> Thanks for pointing it out.
> - For the analysis of in-sentence attacks in Section 5.1, we have added clarification in this revision that the "attention averaging" behavior is of the final layer only with a caveat (in both Section 5.1 and Appendix F.1) that attention is not a reliable proxy for indicating importance over input tokens.
> - For UAT attack in Section 5.2, we have updated our analysis. According to the recommended work [1], gradient\*attention is a better proxy for token importance ranking, as it requires an important token to have both a high attention weight and a high calculated gradient with respect to its attention weight. We use the gradient\*attention based proxy and provide updated visualization for all examples. While we've replaced the interpretation of attention weights with the gradient\*attention based proxy, the observed behavior still holds according to the qualitative studies. The details of the new proxy are attached in Appendix F.2.1. The updated visualization figures can be found in Figure 5 and Appendix F.2.2.
>
> [1.4] In Section 5, there are only several case studies without any quantitative results.
>
> Thanks for the suggestion. We've conducted quantitative studies for UAT to support the observed behavior.
>
> We've designed metrics of corpus-level Degree of Distraction (cDoD) and Recognition of the Essential token (cRoE) to investigate whether the LM steered by robust prefix ignores the distraction from the adversarial trigger tokens and assign the highest importance to the most essential token with UAT-attacked inputs as the baseline does with clean inputs. Evaluating with the two metrics, we find that the scores of our framework with N=3 are better than that of other methods with statistical significance (p < 0.001). This also explains the results shown in Figure 3. The detailed study is attached in Appendix F.2.3.
>
> In general, we believe the analysis in Section 5 is worthwhile (as is also supported by Reviewer rkLx) and tried our best to improve the analysis and presentation so that it's more reliable. We believe the improved analysis section in this revision can bring more insight in understanding the behavior of the robust prefix-tuning framework.
>
> [1.5] Explain why a word attends to itself as well instead of only attending to the contexts in the attention visualization figures.
>
> Thanks for the question. In the visualization figures, each row represents the time step at which the token (labeled on the left) is inputted. At time step $i$, the $i$-th inputted token ($x_i$) attends to itself ($x_i$) as well as all previous tokens ($x_{<i}$). This is because GPT-2 is an autoregressive LM with causal masking, and attention maps are masked into a lower triangle matrix. Here is a reference about how the causal mask is implemented (https://github.com/huggingface/transformers/blob/master/src/transformers/models/gpt2/modeling_gpt2.py#L140-L145) and applied to attention maps (https://github.com/huggingface/transformers/blob/master/src/transformers/models/gpt2/modeling_gpt2.py#L205-L206) in HuggingFace.
>
> [1.6] Improve the presentation in the caption of attention visualization figures.
>
> Thanks. We've improved the caption of the visualization figures with more clarification in this revision.
>
> [1.7] Section 6 is difficult to follow and disconnected from the rest of the paper, the presentation of which should be improved.
>
> Thanks. We've put Section 6 into Appendix G and briefly mentioned it as a remark at the end of the method section.
>
> [1.7] Minor issues in the presentation: Eq 4, an undefined variable X_C suddenly comes in without explanation.
>
> Thanks. The "X_C" is a typo and should be "H_C". We've corrected it in the revision.
>
> [1.8] Minor issues in the presentation: explain Eq (8) with more text given that this is an important equation for the proposed method.
>
> Thanks. We've explained Eq (8) with more text in the revision.
>
> References:
> [1] Sofia Serrano and Noah A. Smith. Is Attention Interpretable? ACL 2019.

---

> > ### Comment · Reviewer_yKwP · 2021-11-27
> > **Thank you for the response.**
> >
> > I think that the paper presentation is improved a lot in the revision. Most importantly, it is great to see the proposed method performs well with small batch sizes, which shows that the method can work in a realistic, online setting. Therefore, I have increased my score and lean to acceptance.

---

> > > ### Author Response · Authors · 2021-11-28
> > > **Thank you for the acknowledgment!**
> > >
> > > Hi,
> > >
> > > Thank you for the acknowledgment as well as the time and effort you spend reviewing our work. We benefited a lot from your suggestions!
> > >
> > > Best regards,
> > >
> > > Authors

---

### Author Response · Authors · 2021-11-23
**Response to all reviewers**

We are grateful to the four anonymous reviewers who offered many insightful and constructive remarks on our work. We benefited a lot from your comments and suggestions!

We tried our best to revise this paper according to the reviewers' comments. To make our efforts clear to the reviewers, we first summarize our modifications here. We have also replied to all reviewers with their comments and our corresponding responses in detail.

The major changes of the revision include:
- Clarify the test batch size used in experiments and conduct experiments under more realistic threat models (Section 4.5 and Appendix E).
- Update the analysis section with a more reliable proxy and provide both qualitative and quantitative studies to support our findings (Section 5 and Appendix F).
- Provide both standard and adversarial full tuning baselines for comparison (Appendix A).

There are also many minor modifications that might improve the presentation. Some of them can be found in the detailed replies.

We would like to summarize the challenges on robustness for prefix-tuning and the contributions of our robust prefix-tuning framework:

- Difficulty in optimization for prefix-tuning. As the number of parameters in a prefix is small, the difficulty in optimizing the prefix might have brought challenges on robustness to prefix-tuning. Table 6 shows that the performance of prefix-tuning baselines is surpassed by that of full-tuning baselines under attack. In contrast, our framework substantially improves the robustness of prefix-tuning under different attacks.

- Trade-off between space and time. Prefix-tuning is lightweight but might take more epochs to converge (due to the optimization difficulty; also see the dev acc curves in Figure 2-(a) for reference). As a result, an ideal solution should be neither harmful to the benefits of prefix-tuning (lightweightness, modularity, not modifying the LM parameters) nor too slow for fear that the time complexity of prefix-tuning further deteriorates. Our framework serves as a solution that keeps the strengths without weakening the weakness.

- Lack of robustness against UAT. Existing defense approaches against UAT maintain additional adversary detectors and are not tested on pretrained LM due to its large scale. To the best of our knowledge, our framework is the first solution to defend against UAT by tuning an additional prefix based on prefix-tuning for pretrained LM. Positioned before the adversarial triggers, the robust prefix regulates the activation at the output position so that "the distraction can be ignored".

The detailed discussion about the challenges and the contributions can be found in Appendix A. We hope the revision and the following responses can address your concerns!

---

### Decision · Program_Chairs · 2022-01-20

**Decision:**

Accept (Poster)

**Comment:**

This paper tackles a relatively novel problem that is the result of recent work on prefix tuning - specifically the need to be robust to adversarial perturbation in the context of prefix tuning and they show a method for achieving this without requiring more storage and obtain good results.

There were some clarity issues that were addressed by the reviewers during the rebuttal. The main issue that was pointed out was the effect of batch size on the success of the model. The authors gave experiments with batch size 1 where results are less impressive but still outperform the baseline. Also the authors say that for now they are not considering the case where only some of the elements in the batch are adversarial, which I think is ok for a research paper on such a cutting-edge topic.

Thus, the result of the discussion is to lean to accept this paper given that it is now more clear, has experiments that make it clear what the benefits are in realistic settings and obtains improvements.